# Functional role of Tet-mediated RNA hydroxymethylcytosine in mouse ES cells and during differentiation

Jie Lan[1,9], Nicholas Rajan[1,9], Martin Bizet[1], Audrey Penning [1], Nitesh K. Singh[1], Diana Guallar [2], Emilie Calonne[1], Andrea Li Greci[1], Elise Bonvin[1], Rachel Deplus[1], Phillip J. Hsu[3], Sigrid Nachtergaele[3], Chengjie Ma[4], Renhua Song [5], Alejandro Fuentes-Iglesias [2], Bouchra Hassabi[1], Pascale Putmans[1], Frédérique Mies[1], Gerben Menschaert[6], Justin J. L. Wong [5], Jianlong Wang [7], Miguel Fidalgo [2], Bifeng Yuan [4] & François Fuks [1,8✉]

Tet-enzyme-mediated 5-hydroxymethylation of cytosines in DNA plays a crucial role in mouse embryonic stem cells (ESCs). In RNA also, 5-hydroxymethylcytosine (5hmC) has recently been evidenced, but its physiological roles are still largely unknown. Here we show the contribution and function of this mark in mouse ESCs and differentiating embryoid bodies. Transcriptome-wide mapping in ESCs reveals hundreds of messenger RNAs marked by 5hmC at sites characterized by a defined unique consensus sequence and particular features. During differentiation a large number of transcripts, including many encoding key pluripotency-related factors (such as Eed and Jarid2), show decreased cytosine hydroxymethylation. Using Tet-knockout ESCs, we find Tet enzymes to be partly responsible for deposition of 5hmC in mRNA. A transcriptome-wide search further reveals mRNA targets to which Tet1 and Tet2 bind, at sites showing a topology similar to that of 5hmC sites. Tet-mediated RNA hydroxymethylation is found to reduce the stability of crucial pluripotency-promoting transcripts. We propose that RNA cytosine 5-hydroxymethylation by Tets is a mark of transcriptome flexibility, inextricably linked to the balance between pluripotency and lineage commitment.

[1] Laboratory of Cancer Epigenetics, Faculty of Medicine, ULB Cancer Research Center (U-CRC), Welbio Investigator, Université Libre de Bruxelles (ULB), Brussels, Belgium. [2] CiMUS, Universidade de Santiago de Compostela–Health Research Institute (IDIS), Santiago de Compostela, Coruña, Spain. [3] Department of Chemistry, Department of Biochemistry and Molecular Biology, Institute for Biophysical Dynamics, and Howard Hughes Medical Institute, University of Chicago, Chicago, IL 60637, USA. [4] Key Laboratory of Analytical Chemistry for Biology and Medicine (Ministry of Education), Department of Chemistry, Wuhan University, 430072 Wuhan, People's Republic of China. [5] Epigenetics and RNA Biology Program Centenary Institute, The University of Sydney, Camperdown, NSW 2050, Australia. [6] Department of Mathematical Modeling, Statistics and Bioinformatics, Faculty of Bioscience Engineering, Lab of Bioinformatics and Computational Genomics, Ghent University, Ghent, Belgium. [7] Department of Medicine, Columbia Center for Human Development (CCHD), Columbia University Irving Medical Center (CUIMC), New York, NY 10032, USA. [8] WELBIO (Walloon Excellence in Lifesciences & Biotechnology), Brussels, Belgium. [9] These authors contributed equally: Jie Lan, Nicholas Rajan. ✉email: ffuks@ulb.ac.be

n DNA, the family of TET methyldioxygenases (TET1, TET2, and TET3) is known to catalyze hydroxylation of 5-methylcytosine to generate 5-hydroxymethylcytosine[1,2]. This reaction, which requires $Fe^{2+}$ and α-ketoglutarate as co-factors, adds an additional layer of complexity to the epigenetic regulation of DNA methylation, as it can act as an intermediate in DNA demethylation pathways[2–4]. Recent advances have provided a more precise picture of the roles of TET-mediated DNA hydroxymethylation in several diseases such as cancer[5–9] and in various biological contexts[10–12]. Notably, it is increasingly clear that DNA hydroxymethylation has a role in key physiological processes, including pre-implantation[13–15], ESC pluripotency, and differentiation[16–22]. For example, TET triple knockout (TKO) and single TET KO studies reveal that while TET proteins are not required for ESC maintenance, they are essential for the proper differentiation capacity and the generation of functional embryonic structures[20,21,23].

Previous work showed that Tet-mediated 5-hydroxymethylation (5hmC) occurs also in RNA context[24–30], but in RNA its roles are just beginning to be appreciated. Tet-deficient *Drosophila* fruitflies suffer impaired brain development, accompanied by decreased RNA hydroxymethylation[28]. In mammals, Tet2 acts via 5hmC marking of RNA to promote pathogen infection-induced myelopoiesis through mRNA oxidation[29] and to control endogenous retroviruses (ERVs)[30]. Tet-mediated RNA hydroxymethylation has also been reported to occur in ESCs[24]. However, to date, the distribution, function, and biological relevance of 5hmC remain unknown.

Here, we show that Tet enzymes are required for deposition of 5hmC in mRNAs, and notably in key pluripotency-related transcripts. Interestingly, we find that during differentiation, a large number of these transcripts have a reduced level of cytosine hydroxymethylation. We report that 5hmC reduces the stability of important pluripotency-promoting transcripts, and propose Tet-mediated RNA hydroxymethylation as an additional level of regulation of the ESC self-renewal network.

## Results

### Transcriptome-wide distribution of 5hmC in ESCs and EBs.
We initiated our study by assessing the transcriptome-wide 5hmC landscape in mouse ESCs. For this we used our previously described hMeRIP-Seq method[28], involving immunoprecipitation of 5hmC-containing RNA with an anti-5hmC antibody (Fig. 1a and Supplementary Fig. 1a), followed by next-generation sequencing (see Methods for experimental details and validation of antibody specificity). This approach revealed a total of 1633 peaks ($q$-values < 0.05) in 795 transcripts (Fig. 1b, c and Supplementary Data 1). The top hydroxymethylated mRNA targets are shown in Supplementary Fig. 1b and examples of enrichment profiles with the corresponding input tracks are shown in Fig. 1c (additional examples are shown in Supplementary Fig. 1c). We observed a non-random distribution of 5hmC, the mark occurring mostly in introns (Fig. 1d). Subsequent analyses revealed a specific UC-rich motif at peak centers (Fig. 1e and Supplementary Fig. 1d), consistently with findings of our previous study on *Drosophila* S2 cells[28].

To see whether transcripts relevant to ESC pluripotency might be present among those identified here, we compared the above-mentioned hMeRIP-Seq data sets with publicly available mouse ESC data sets defining signatures for the regulatory circuitry controlling the embryonic stem cell state[31–35]. Of the 795 5hmC-modified transcripts identified, 110 were found to encode pluripotency-related factors, including key ESC pluripotency regulators such as Eed, Jarid2, Smarcc1, Paf1, and Mbd3 (Fig. 1f and Supplementary Fig. 1e and Supplementary Data 1). We

observed the same features of non-random distribution of 5hmC peaks within these pluripotency-related transcripts, with the mark occurring mostly in introns (Supplementary Fig. 1f). This transcriptome-wide assessment of 5hmC in WT mouse ESCs thus highlights a unique distribution and features of 5hmC sites in hundreds of transcripts, notably of many key pluripotency-related mRNAs.

The above-mentioned strong 5hmC enrichment within introns (cf. Fig. 1d and Supplementary Fig. 1f) prompted us to assess the level of 5hmC by dot blotting on the three following RNA fractions: nascent chromatin-associated, nucleoplasmic, and cytoplasmic. As shown in Supplementary Fig. 1g, we observed that chromatin-associated RNAs were readily hydroxymethylated. These data suggest enrichment in 5hmC of intronic regions of unspliced nascent pre-mRNAs. We also evaluated the role of Tet-mediated hydroxymethylation in splicing regulation, by means of paired-end RNA-Seq in WT and TKO ESCs, followed by differential splicing analysis. We found Tet-mediated hydroxymethylation to be associated with a higher ratio of spliced to unspliced transcripts (Supplementary Fig. 1h, i and Supplementary Data 2).

We next examined how the level and distribution of 5hmC might change during mESC differentiation to embryoid bodies (EBs). We used conditions for spontaneous differentiation of ESCs to EBs at an early time (day 4), which allowed us to focus on the role played by Tet1 and Tet2 at an early stage of ESC differentiation. In agreement with a previous report[36], transcript-level expression of *Tet1* and *Tet2*, as measured by RT-qPCR, were decreased upon ESC-to-EB differentiation, while *Tet3* was still barely expressed (Fig. 1g). Proper differentiation of ESCs to EBs was checked by quantifying markers of pluripotency and early differentiation (Supplementary Fig. 1j). We first assessed the global 5hmC level by dot blotting applied to RNA extracts. EBs displayed a lower 5hmC signal than ESCs (Fig. 1h and Supplementary Fig. 1k). We then performed hMeRIP-Seq on ESCs and EBs. As shown in Fig. 1i, 5hmC marking was found to decrease in over 80% of the transcripts upon ESC-to-EB differentiation. The observed 5hmC changes were widely distributed within transcripts (Supplementary Fig. 1l and Supplementary Data 3). Of the 649 mRNAs showing reduced 5hmC in EBs vs ESCs, 72 encode pluripotency-promoting factors, e.g., Eed, Jarid2, and Dab1 (Fig. 1i, j and Supplementary Fig. 1m). ESC-to-EB differentiation thus leads, concomitantly with reduced *Tet1* and *Tet2* expression, to a marked decrease in 5hmC, notably affecting key pluripotency-related mRNAs.

### Tet-mediated hydroxymethylation in ESCs.
What is the contribution of Tet proteins to mRNA hydroxymethylation in ESCs? To answer this question, we used previously generated *Tet1/2/3* triple knockout (TKO) mouse ESCs[21]. In line with the previous work[24], TKO ESCs showed a substantially lower (~50% lower) global 5hmC level than WT ESCs, as measured by dot blotting and mass spectrometry (Fig. 2a and Supplementary Fig. 2a). It is noteworthy that m5C remained at a similar level in TKO cells (Supplementary Fig. 2b), consistently with previously published data. Since Vitamin C is a known cofactor for Tet-mediated DNA hydroxymethylation in ESCs[4] (Supplementary Fig. 2c), we tested whether Vitamin C might also induce Tet-dependent RNA hydroxymethylation. This proved to be the case: dot blotting applied to RNA from WT and TKO ESCs revealed, upon Vitamin C treatment, a rise in the global level of 5hmC in the WT cells only (Supplementary Fig. 2d).

To see which Tets might be responsible for 5hmC marking, we first measured the global 5hmC level in *Tet1/2/3* triple knockout (TKO)[21], *Tet1/2* double knockout (DKO)[20], and *Tet3* knockout

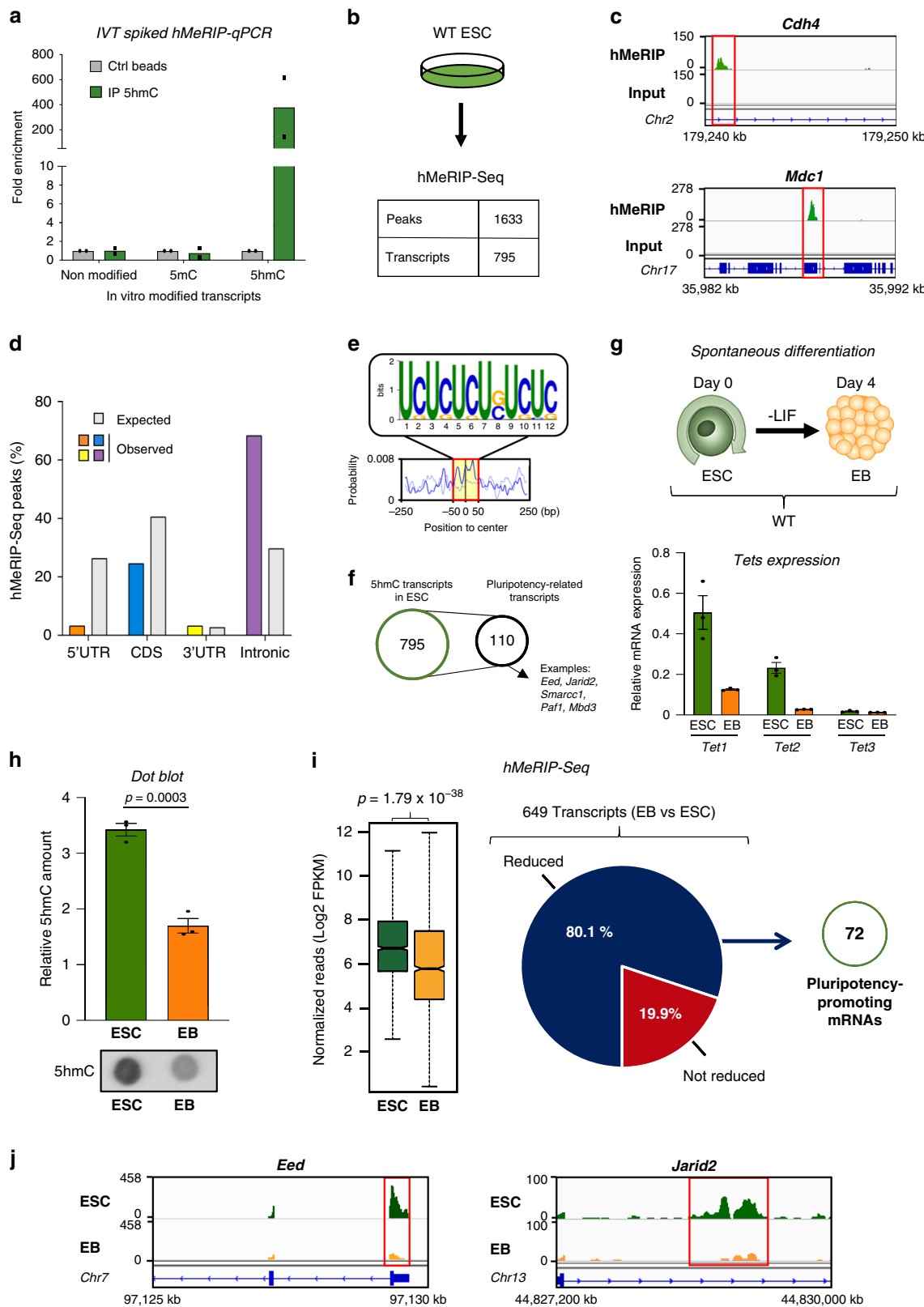

ESCs[37]. Dot blots for *Tet1-* and *Tet2*-depleted ESCs (DKO) showed a reduction of the 5hmC level similar to that observed for TKO ESCs, while dot blots for *Tet3* KO ESCs showed no decrease in global 5hmC (Supplementary Fig. 2a). These results indicate that while Tet1, Tet2, or both are involved in 5hmC marking of mRNAs in ESCs, this seems not to be the case for Tet3.

To examine the contribution of Tets to 5hmC marking at the transcript level, we performed hMeRIP-Seq on WT and TKO ESCs. As shown in Fig. 2b, we observed a significant reduction of 5hmC reads ($P < 10^{-34}$) in TKO as compared to WT ESCs. In Tet-depleted ESCs, 68.1% of the mRNAs (575 transcripts) showed a reduced 5hmC level (Supplementary Data 4), among

**Fig. 1 Transcriptome-wide distribution of 5hmC in ESCs and EBs. a** Specificity of 5hmC antibody. Only the 5hmC-modified transcript shows enrichment after hMeRIP compared to the controls. Unmodified, 5mC-modified, and 5hmC-modified transcripts (IVT: in vitro transcribed) were used to spike total RNA prior to hMeRIP-qPCR. Data are means ± SEM ($n = 2$ independent experiments). **b** hMeRIP-Seq in WT ESCs reveals the presence of 5hmC within many transcripts ($n = 3$). Experiments were performed in biological triplicate and results were normalized as described in the "Methods" section. **c** Exemplative hMeRIP-Seq profiles of *Cdh4* and *Mdc1* in WT ESCs with their corresponding input control tracks (IGV tracks) (red frame shows peak location). **d** Bar chart showing the distribution of 5hmC peaks according to the type of structural element within transcripts, next to the expected distribution. **e** Top sequence motif identified in the centers of 5hmC peaks (E-value < 2.2e−117). **f** 5hmC is found in many key pluripotency-related mRNAs. Comparison of the above hMeRIP-Seq data sets with publicly available mouse ESC data sets[10–14], with representative examples of known transcripts encoding ESC core pluripotency regulators such as *Eed, Jarid2, Smarcc1, Paf1*, and *Mbd3*. **g** Scheme of the previously reported protocol[36] used for spontaneous differentiation of ESCs into EBs upon LIF removal, as described in the Methods section, with their relative Tet expression levels as measured by RT-qPCR. Data are means ± SEM ($n = 3$ independent experiments). **h** Decreased global 5hmC during spontaneous differentiation, as assessed by dot blotting. Data are means ± SEM (representative blot from three independent experiments, two-tailed Student's *t*-test). **i** Many pluripotency-related transcripts show reduced 5hmC during differentiation. Left: Box plot for hMeRIP-Seq ESCs and EBs ($n = 3$ independent experiments, two-tailed Student's *t*-test). In the box plot, the boxes represent the interquartile range of the records, and the lines across the boxes indicate the median value of the records. The whiskers indicate the highest and lowest values among the records that are no more than 1.5 times greater than the interquartile range. The range between notches represents the 95% confidence interval. Right: Pie chart highlighting the percentage of transcripts showing reduced 5hmC marking, 72 of which are identified as pluripotency-promoting mRNAs. **j** Exemplative hMeRIP-Seq profiles of *Eed* and *Jarid2* in ESCs vs EBs (IGV tracks) (red frame shows peak location). Source data are provided as a Source Data File.

which 52 are known to encode critical pluripotency factors such as Eed, Dab1, and Sfpq (Fig. 2c, d and Supplementary Fig. 2e). In mRNAs showing a reduced 5hmC level in TKO ESCs, we found an overrepresented UC-rich motif (Fig. 2e) highly similar to the 5hmC site identified in WT ESCs (cf. Fig. 1e). Furthermore, the 5hmC decrease was found to occur mostly in introns (Fig. 2f). Together, these results indicate that Tets are required for deposition of 5hmC in mRNAs, and notably in key pluripotency-related transcripts. Given that Tets knockout did not totally abolish 5hmC marking either globally or at the transcript level, it seems likely that additional enzymes and/or chemical processes contribute to mRNA hydroxymethylation in ESCs. To test whether 5hmC in mRNA might form through other chemical processes, we specifically evaluated whether 5hmC might be induced by cellular reactive oxygen species (ROS). This seems not to be the case, as treatment of ESCs with either buthionine sulfoximine (BSO) or $H_2O_2$ did not change the global level of 5hmC in mRNA (Supplementary Fig. 2f).

**Tet1- and Tet2-bound mRNAs in ESCs.** As the mRNA targets of Tets are unknown, we next sought to identify these targets throughout the transcriptome by generating CRISPR knock-in ESCs for *Tet1* and *Tet2*. Using the CRISPR genome-editing tool in ESCs, we added a Flag-tag to endogenous Tet1 or Tet2. RNA immunoprecipitation (RIP) with anti-Flag antibody was then performed, followed by deep sequencing (Fig. 3a and Supplementary Fig. 3a). As shown in Fig. 3b, RIP-Seq for endogenous Tet1 identified 7798 bound targets. Similar experiments for Tet2 revealed its binding to 6659 transcripts (Fig. 3c and Supplementary Data 5). Interestingly, an RNA-binding domain (RBD) within Tet2 has recently been identified by proteomic approach and is a sequence adjacent to the C-terminal catalytic domain[38]. Exploiting this finding, we used CRISPR-Cas9 to delete from endogenous Tet2 the 54 amino acids corresponding to the whole sequence encoding the identified site (Fig. 3a and Supplementary Fig. 3a). The corresponding knock-in cells thus produced a Flag-tagged Tet2 protein, either WT or deleted of the RBD (Tet2ΔRBD). RIP-Seq for Tet2ΔRBD revealed that about 30% of Tet2 targets are dependent on its RBD (Fig. 3c–e). The identified RBD thus contributes at least partly to specific Tet2 targeting. A comparison of the Tet1- and Tet2-RIP-Seq data revealed considerable and significant overlap between Tet2- and Tet1-bound targets, corresponding to 78.7% of the Tet2-bound transcripts (Fig. 3f). We also compared our RIP-Seq data for Tet1 and Tet2

with the hMeRIP-Seq data. Although many Tet1- and Tet2-interacting transcripts seemed not to be hydroxymethylated, 64.5% of the identified hydroxymethylation targets appeared to interact with Tet1 and/or Tet2 (Fig. 3g). We found that when Tet1 and Tet2 are bound to 5hmC targets, they are mostly bound together, rather than alone (Fig. 3g). This suggests that both Tet1 and Tet2 contribute to 5hmC and that they have redundant roles in mRNA hydroxymethylation in ESCs. We further found Tet1 and Tet2, like 5hmC, to associate preferentially with intronic regions (Supplementary Fig. 3b). Likewise, within 5hmC-enriched sites, Tet1 and Tet2 appeared to bind targets preferentially characterized by a UC-rich motif (Supplementary Fig. 3c). Interestingly, the percentage of pluripotency-related transcripts showing both enrichment in 5hmC and binding to Tet1 and/or Tet2 was particularly high, i.e., 70% (Supplementary Fig. 3d). These transcripts notably included *Eed, Jarid2, Smarcc1*, and *Dab1*. It is worth mentioning that in addition to binding to 5hmC-modified targets, Tet1 and Tet2 also bound to many unmodified transcripts. Using publicly available data[19] we observed, upon comparing Tet1/2-bound 5hmC-modified and unmodified RNAs, a lower level of 5-methylcytosine in genes bodies related to unmodified RNAs than in genes related to 5hmC-modified ones (Supplementary Fig. 3e). This suggests potential catalysis-independent roles for Tet1 and Tet2.

To further investigate the effect of Tet1/2 binding on 5hmC-modified and unmodified mRNAs, we performed RNA-Seq experiments on TKO ESCs and analyzed upregulated and downregulated transcripts upon Tet depletion. Firstly, by comparing 5hmC targets from hMeRIP-Seq with Tet-regulated transcripts, we found 55.6% of the 5hmC-enriched targets to be upregulated and 44.4% to be downregulated (Supplementary Fig. 3f). Secondly, a comparison of Tet1/Tet2-bound mRNAs from RIP-Seq with RNA-Seq in TKO ESCs showed both upregulated (65.9%) and downregulated transcripts (34.1%) (Supplementary Fig. 3g). Lastly, we also looked at the overlap between 5hmC-enriched targets bound by Tet1/2 and up- or downregulated transcripts. We found a significant number of downregulated transcripts harboring 5hmC to be bound by Tet1/2 (68.2%). Many upregulated transcripts enriched in 5hmC were also found to interact with Tet1/2 (67.3%) (Supplementary Fig. 3h).

This transcriptome-wide investigation thus shows that a large number of transcripts are bound by Tet1, Tet2, or both. We observed that the majority of 5hmC targets are bound by Tet1/2,

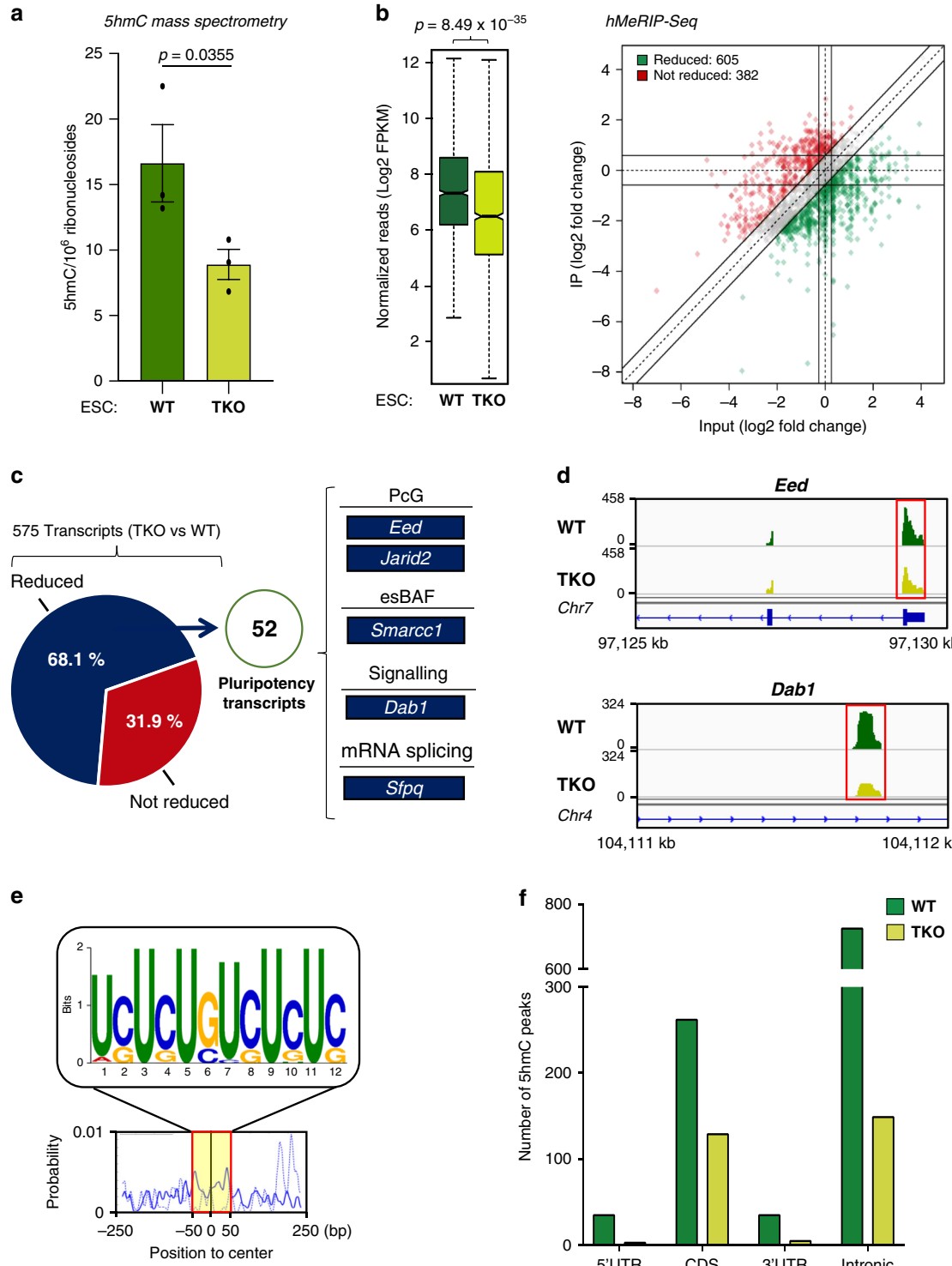

**Fig. 2 Tets are required for 5hmC in ESCs, notably of key pluripotency-related mRNAs. a** Reduced global 5hmC in TKO ESCs, as measured by mass spectrometry. Data are means ± SEM ($n = 3$ independent experiments, one-tailed Student's $t$-test). Source data are provided as a Source Data File. **b** Decreased 5hmC at many peaks in TKO ESCs. Left: Box plot showing a significant difference in the normalized number of 5hmC reads between WT and TKO ESCs ($n = 3$ independent experiments, one-tailed Student's $t$-test). In the box plot, the boxes represent the interquartile range of the records, and the lines across the boxes indicate the median value of the records. The whiskers indicate the highest and lowest values among the records that are no more than 1.5 times greater than the interquartile range. The range between notches represents the 95% confidence interval. Right: Quadrant chart showing differential 5hmC peaks in WT vs TKO ESCs. **c** Tet-mediated 5hmC marking of core pluripotency transcripts. Pie chart highlighting the percentage of transcripts whose 5hmC marking appears reduced, 52 of which are known to be involved in pluripotency. **d** Exemplative hMeRIP-Seq profiles of *Eed* and *Dab1* in TKO vs WT ESCs (IGV tracks) (red frame shows peak location). **e** Top sequence motif identified at the centers of 5hmC peaks reduced in WT vs TKO ESCs ($E$-value < 3.9e−086). **f** Non-random distribution of Tet-mediated 5hmC marking. Bar chart showing, in WT and TKO ESCs, distinct distributions of 5hmC peaks among types of structural elements within transcripts.

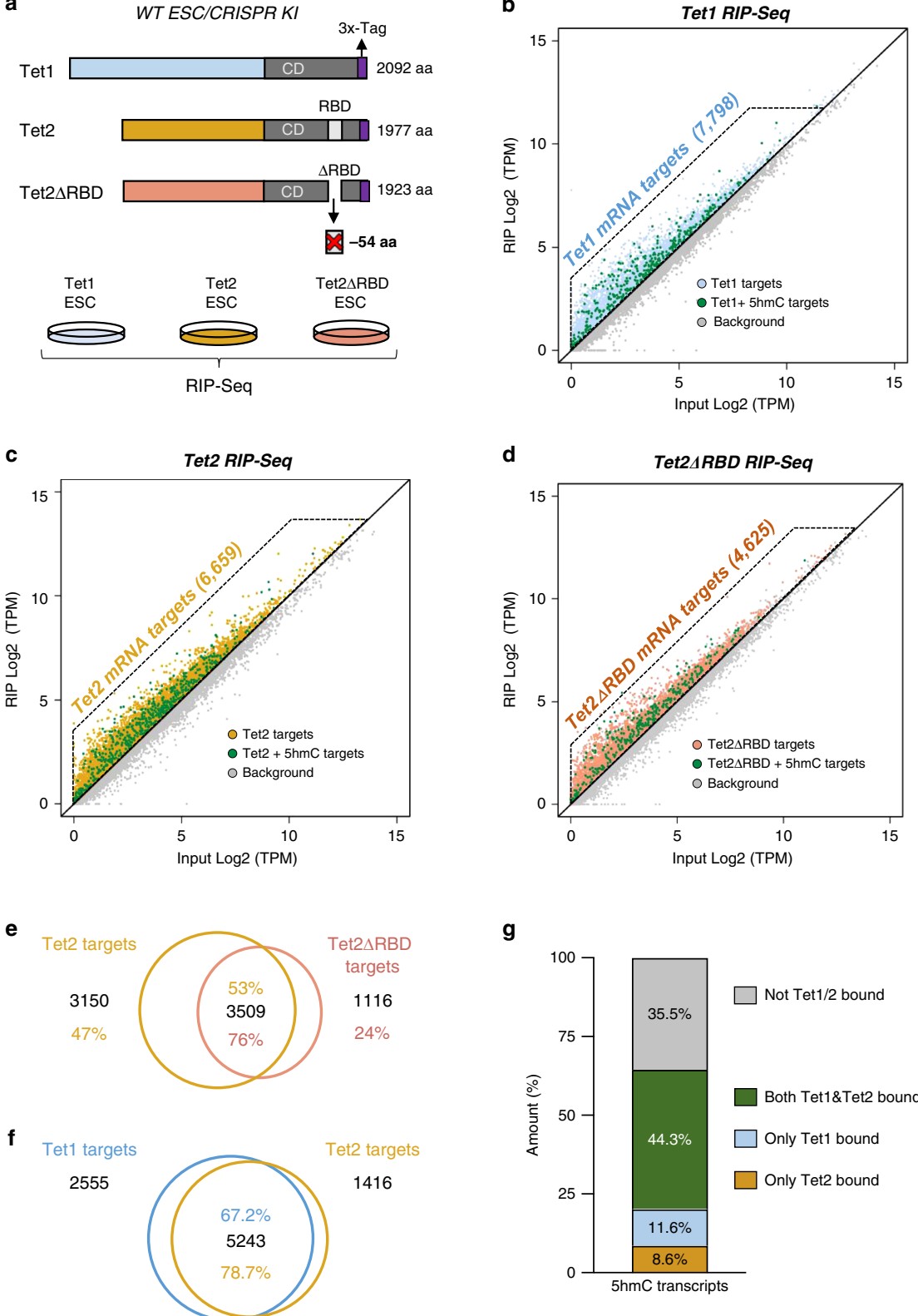

**Fig. 3 Tet1 and Tet2 bind to specific mRNAs within the transcriptome. a** Diagram illustrating CRISPR-mediated tagging of endogenous Tet1, Tet2, and Tet2ΔRBD proteins (CRISPR KI) and RIP-Seq experiment design. **b** Many mRNAs are bound by Tet1. Scatterplot showing the log-fold enrichment over input after Tet1-bound RNA immunoprecipitation. **c** RNA-binding targets of Tet2. Scatterplot showing the log-fold enrichment over input after Tet2-bound RNA immunoprecipitation. **d** Tet2 RBD is involved in mRNA targeting. Scatterplot showing the log-fold enrichment over input after Tet2ΔRBD-bound RNA immunoprecipitation. **e** Tet2 binding to many RNAs depends on Tet2 RBD. Venn diagram showing the overlap between Tet2- and Tet2ΔRBD-bound RNA targets identified by RIP-Seq. **f** Many common Tet1 and Tet2 targets. Venn diagram showing the overlap between Tet1- and Tet2-bound RNA targets identified by RIP-Seq. **g** 5hmC targets are often bound by both Tet1 and Tet2. Stacked bar chart showing the overlap between 5hmC-containing transcripts and RNA targets bound by both Tet1 and Tet2, only Tet1, or only Tet2. All RIP-Seq were performed at least in biological duplicate.

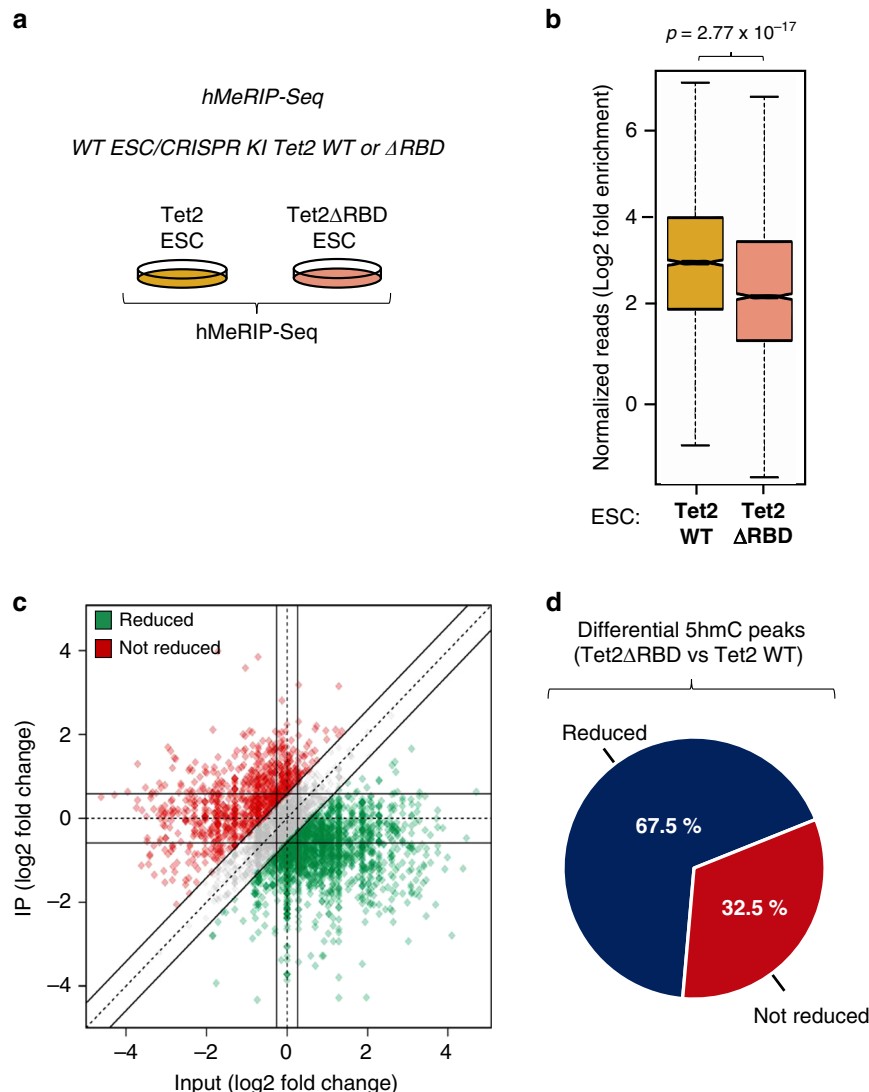

**Fig. 4 Tet2-mediated RNA hydroxymethylation depends, at least in part, on Tet2 RBD. a** Scheme illustrating the hMeRIP experimental design using CRISPR Tet2WT KI and CRISPR ΔRBD KI ESCs. **b** Box plot showing a significant difference in the normalized number of 5hmC reads between Tet2WT and Tet2ΔRBD ESCs (*n* = 3 independent experiments, one-tailed Student's *t*-test). In the box plot, the boxes represent the interquartile range of the records, and the lines across the boxes indicate the median value of the records. The whiskers indicate the highest and lowest values among the records that are no more than 1.5 times greater than the interquartile range. The range between notches represents the 95% confidence interval. **c** Quadrant chart showing differential 5hmC peaks in CRISPR Tet2WT KI vs CRISPR Tet2ΔRBD KI ESCs. **d** Pie chart highlighting the percentage of transcripts whose 5hmC level is reduced in Tet2ΔRBD- compared to Tet2WT-producing cells.

among which many pluripotency-related transcripts, and that this interaction is characterized by a defined consensus site and topology.

**Tet2-mediated RNA hydroxymethylation depends partially on its RBD.** Having found that the Tet2 RBD contributes to Tet2 targeting and binding to transcripts (cf. Fig. 3c–e), we evaluated to what extent this domain is required for Tet2-mediated RNA hydroxymethylation. To this end, we performed hMeRIP-Seq with CRISPR/Cas9 knock-in ESCs for Tet2WT and TET2ΔRBD (Fig. 4a). As depicted in Fig. 4b–d (and Supplementary Data 6), we observed a significant decrease (67.5%) in 5hmC-enriched regions upon the deletion of Tet2 RBD. This shows that Tet2, at least via its RBD, contributes to hydroxymethylation of mRNAs. This is in line with our recent work showing Tet2-mediated RNA hydroxymethylation of endogenous retroviruses[30].

**Tet-deposited 5hmC decreases mRNA stability, notably of core pluripotency transcripts.** What might be the function of Tet1/2-mediated mRNA hydroxymethylation in ESCs? To answer this question, we first investigated whether 5hmC marking might correlate with transcript abundance. The identified 5hmC-modified transcripts were thus ranked according to their abundance. Most 5hmC-modified transcripts appeared in the middle parts of transcript abundance (Supplementary Fig. 4). This preference for transcripts showing medium abundance suggests that 5hmC is not simply a random modification occurring on abundant transcripts. We then wondered whether 5hmC-marked transcripts might differ from unmodified transcripts at the level of translation or RNA decay. To investigate this, we examined published genome-wide data sets for mESCs. The possible effect of 5hmC on translational efficiency was investigated by means of previously reported ribosome profiling (Ribo-Seq) data sets from WT ESC[39]. As shown in Fig. 5a, 5hmC-modified and unmodified

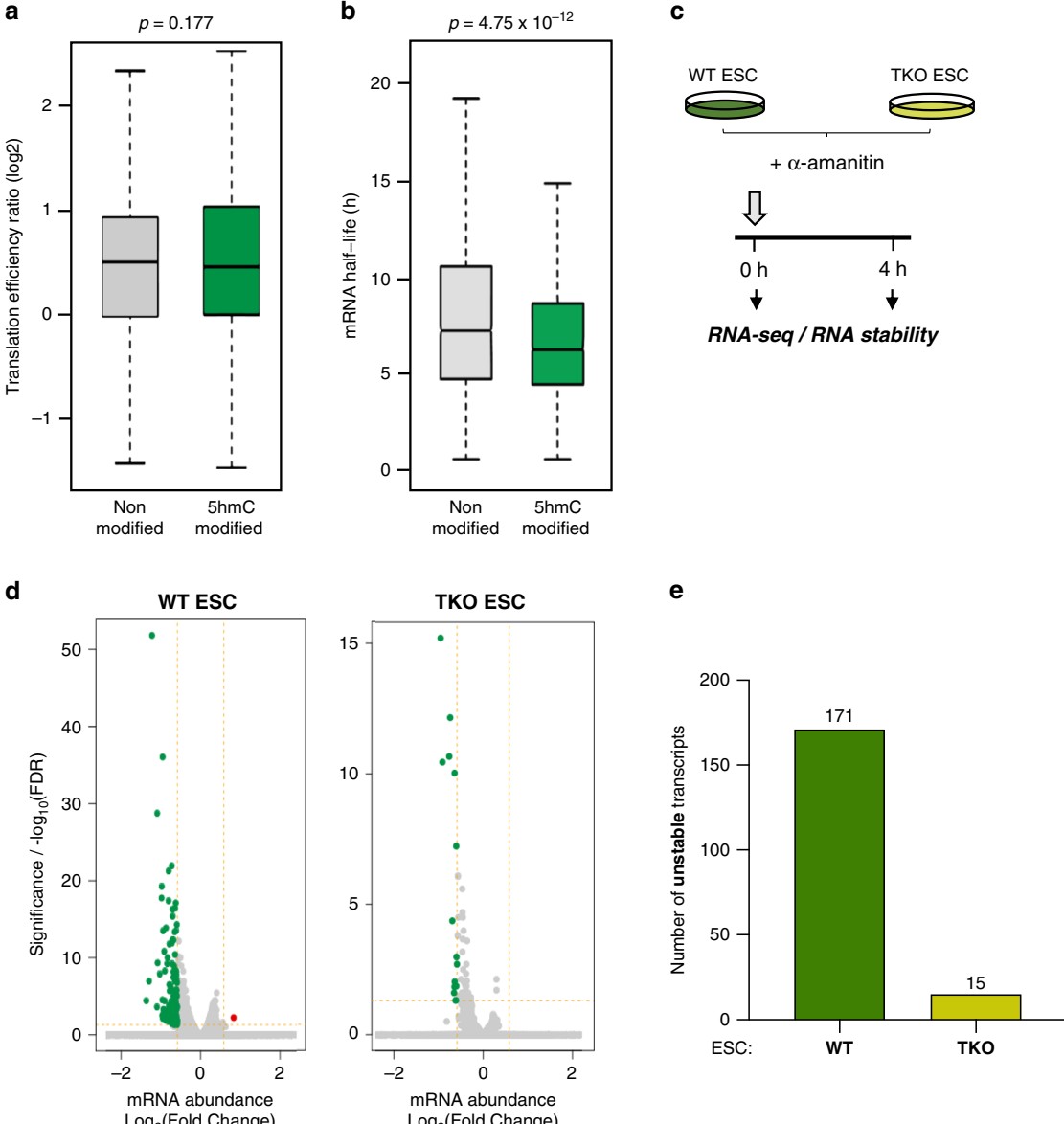

**Fig. 5 Transcriptome-wide analysis of RNA stability in WT vs TKO ESCs. a** 5hmC does not seem to impact mRNA translation. Box plot showing the translation efficiency of non-modified and 5hmC-modified transcripts in WT ESCs using our hMeRIP-Seq data ($n = 3$) and published data[39] (two-tailed Wilcoxon rank-sum test, $P > 0.05$). **b** 5hmC leads to a shorter mRNA half-life. Box plot showing the difference in mRNA half-life between non-modified and 5hmC-modified transcripts in WT ESCs using our hMeRIP-Seq data ($n = 3$) and published data[40] (two-tailed Wilcoxon rank-sum test, $P < 10^{-12}$). In the box plot, the boxes represent the interquartile range of the records, and the lines across the boxes indicate the median value of the records. The whiskers indicate the highest and lowest values among the records that are no more than 1.5 times greater than the interquartile range. **c** Scheme illustrating the protocol for transcription inhibition with α-amanitin in WT and TKO ESCs followed by RNA-Seq. Experiments were performed in biological duplicate. **d** Volcano plots show longer mRNA half-lives in Tet-depleted than WT ESCs. (*P*-value corrected for multi-testing < 0.05 and FC > 1.5). **e** Bart chart showing a greater number of unstable transcripts in WT vs TKO ESCs, as determined by RNA-Seq at 0 and 4 h post α-amanitin treatment.

transcripts showed no difference in translation efficiency. We then examined whether 5hmC might be associated with mRNA stability by analyzing a published data set for mRNA half-life in ESCs[40]. As depicted in Fig. 5b, 5hmC-marked transcripts displayed a significantly shorter mRNA half-life than unmodified transcripts ($P < 10^{-12}$). These results suggest that 5hmC is a chemical mark associated with transcript turnover.

To confirm the effect of 5hmC deposition on transcript stability, we added α-amanitin to WT and TKO ESCs to inhibit transcription and performed RNA-Seq (Fig. 5c). As depicted in Fig. 5d, e (and Supplementary Data 7), we observed longer mRNA half-lives upon Tet depletion in TKO vs WT ESCs. These

results suggest a role for Tet-mediated hydroxymethylation in mRNA stability.

To further probe the contribution of 5hmC in transcript stability, we produced unmodified and 5hmC-modified transcripts by in vitro transcription in the presence of C or 5hmC nucleotides and used them to transfect WT ESCs. Their abundance was measured 6 h and 24 h post-transfection in order to evaluate their relative stability (Fig. 6a). We observed after ESC transfection that in vitro 5hmC-modified transcripts were less stable than their unmodified counterparts (Fig. 6b). These in vitro data are in good agreement with our above data showing that Tet-mediated 5hmC favors fast turnover of RNA transcripts.

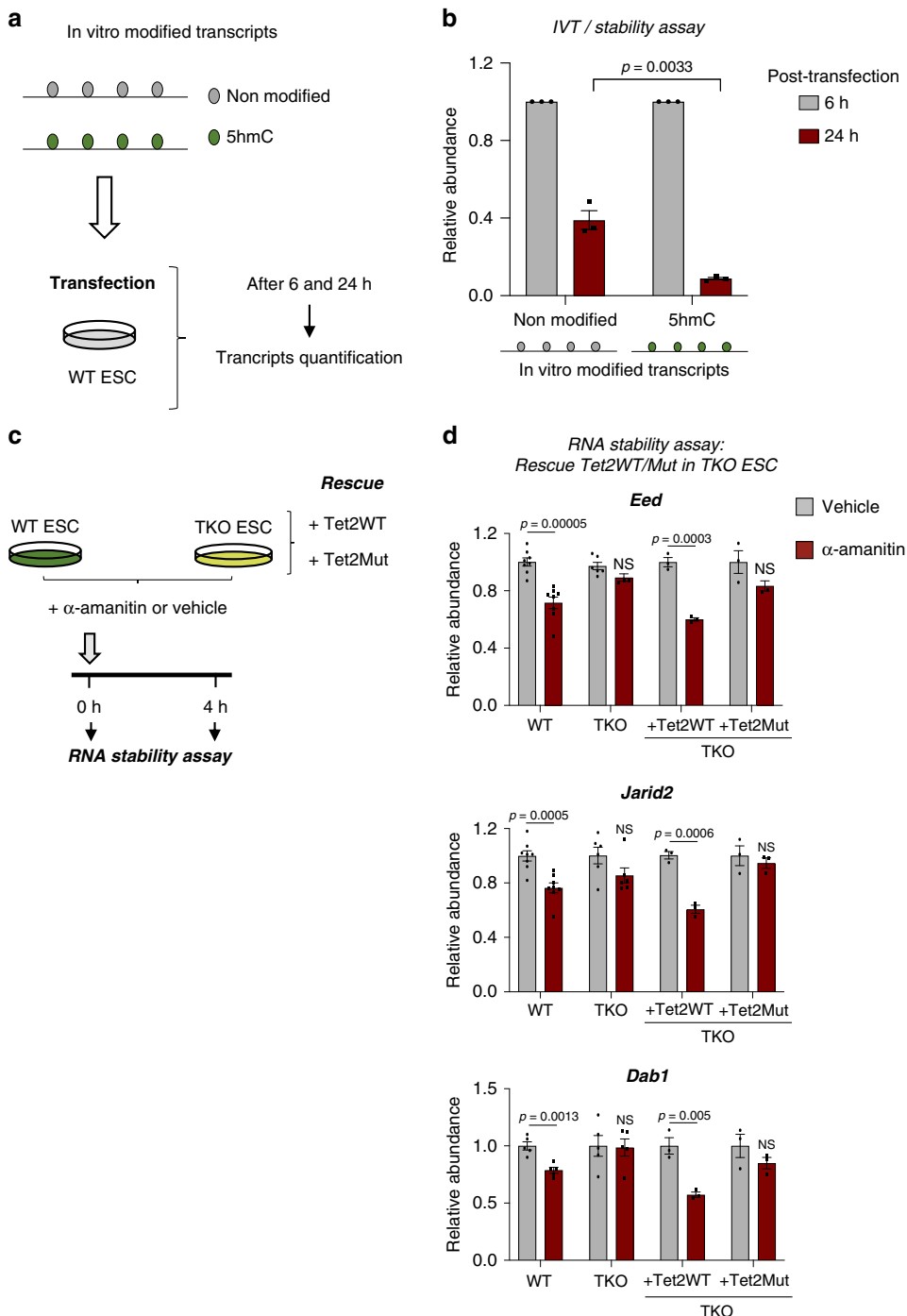

**Fig. 6 Tet-mediated 5hmC reduces the stability of core pluripotency transcripts. a** Scheme illustrating the protocol used to quantify in vitro unmodified and 5hmC-modified transcript abundance after ESC transfection. **b** In vitro 5hmC-modified transcripts are less stable than their unmodified counterparts. Unmodified and 5hmC-modified transcript abundances were measured by RT-qPCR at 6 and 24 h post-transfection. RNA levels were normalized to endogenous *Gapdh* and relative to the transcript level at 6 h post-transfection. Data are means ± SEM ($n = 3$ independent experiments, two-tailed Student's *t*-test). **c** Scheme illustrating the protocol used to inhibit transcription with α-amanitin in WT and TKO ESCs before performing stability assays to detect rescue of the destabilization process by Tet2WT or its catalytic mutant (Tet2Mut). **d** Tet2WT but not Tet2Mut restores destabilization of *Eed*, *Jarid2*, and *Dab1* transcripts in TKO ESCs. 18S rRNA was used as an internal calibrator. Error bars indicate ± SEM for at least three independent experiments. (Two-tailed Student's *t*-test; NS not significant). Source data are provided as a Source Data File.

To validate our findings in vivo, we added α-amanitin to WT and TKO ESCs and monitored, by qPCR, levels of key pluripotency-related mRNAs over a 4 h treatment period. To confirm the involvement of Tet proteins in hydroxymethylation of pluripotency-related mRNAs and the transcript-destabilizing effect of hydroxymethylation, we performed rescue experiments on TKO ESCs with Tet2WT or a Tet2 catalytic mutant (Tet2Mut) (Fig. 6c). As shown in Fig. 6d, *Eed, Jarid2*, and *Dab1* transcripts were significantly less stable in WT ESCs than in TKO ESCs. Moreover, we found wild-type Tet2, but not Tet2Mut, to rescue

the mechanism that destabilizes these transcripts in WT ESCs. An additional example and the *18S* RNA control are shown in Supplementary Fig. 5a.

Finally, we extended our observations on the above pluripotency-related mRNA stability assay by using CRISPR/Cas9 knock-in ESCs for Tet2WT or Tet2ΔRBD. We found Tet2WT, but not Tet2ΔRBD, to decrease the abundance of pluripotency transcripts (Supplementary Fig. 5b).

Overall, these results suggest that 5hmC deposition on pluripotency-related transcripts facilitates their degradation, which depends, at least in part, on Tet2 catalytic activity and on the RBD.

## Discussion

Here we provide evidence of an additional level of regulation of the ESC self-renewal network: RNA hydroxymethylation by Tet enzymes. Our data support a stepwise working model whereby 5hmC mRNA modification acts as an essential regulatory layer to safeguard efficient, timely, authentic downregulation of lineage-specific genes. In this manner, 5hmC can promote a fast response to external cues during cell differentiation (Fig. 7). Specifically, it is well known that a gene expression program in ESCs allows them to self-renew, yet they remain poised to differentiate into all cell types in response to developmental cues. For this, key cell fate determinants need to be expressed to appropriate levels, ensuring that lineage-specific genes are adequately repressed, thus ensuring orderly differentiation of ESC[41]. For example, should pluripotency factors be too highly expressed, this would lead to strong silencing of lineage-commitment genes, with cells remaining in the pluripotent state. On the basis of our data, we propose a model whereby 5hmC marks key ESC fate determinants to limit their levels and ensure their continuous degradation. Concretely, 5hmC would contribute to controlling the abundance of pluripotency-associated factors (such as *Eed* or *Jarid2*), so that they are expressed at appropriate levels (not too high, not too low). This would ensure adequate repression of lineage-specific factors and critically prepare ESCs to rapidly respond to differentiation stimuli (Fig. 7).

Along with the well-described role of DNA hydroxymethylation by Tets in ES cells[19,42,43], our present findings must now be taken into account if one is to understand fully the functional importance of Tets in pluripotency and lineage commitment. Future work should address how Tet enzymes discriminate between DNA and RNA substrates for hydroxymethylation. Elements that might guide Tets to specific substrates include Tet-interacting proteins[30,42], protein *O*-glycosylation marks[44,45], secondary structure, and structural changes, among other possibilities.

Regarding which Tets are responsible, in ESCs, for 5hmC marking of RNAs, our study suggests that Tet3 is not involved but that both Tet1 and Tet2 contribute similarly to RNA m5C oxidation and have redundant functions. Future analyses will be needed to decipher the mechanisms through which Tet1 and Tet2 can substitute for one another in RNA m5C oxidation. Our findings do not exclude the involvement of Tet3 in other cell contexts. Worth adding is our observation that Tet enzymes are only partly responsible for depositing 5hmC in mRNA, consistently with earlier reports[24]. Although we cannot exclude the possibility that other chemical processes besides ROS-related ones might be involved, it could be that enzymes other than Tets deposit 5hmC on RNA. Such enzymes would probably belong, like the Tet proteins, to the family of ferrous-ion- and α-KG-dependent dioxygenases ($Fe^{2+}$ and 2-OG). Further study is warranted to identify additional RNA hydroxymethyltransferases.

An important finding of the present work is the identified transcriptome-wide catalog of Tet1- and Tet2-bound mRNAs. It appears that the majority of 5hmC-modified mRNAs are bound by Tet1 and Tet2, at a defined consensus site with a defined topology. We further show that a recently identified Tet2 RNA-binding domain[38] is crucial for Tet2 targeting to specific transcripts and for their subsequent hydroxymethylation. An RNA-based targeting and oxidation mechanism of this type appears distinct from the reported recruitment of Tet2 to chromatin via the RNA-binding protein Pspc1[30]. The set of Tet-interacting transcripts identified here might constitute an additional class of RNA regulons[46]. It is worth noting that in addition to their binding to 5hmC-modified targets, Tet1 and Tet2 bind also to many unmodified transcripts. To us, this suggests the interesting possibility that besides hydroxylating mRNAs, Tet1 and Tet2 might also function independently of their catalytic activity. Such an "RNA-hydroxymethylation-independent" role would be analogous to the well-described non-catalytic action of Tet1 and Tet2 on DNA, in which Tet proteins associate with diverse chromatin-related machineries such as HDAC and SET1/COMPASS, involved in transcriptional repression or activation[10]. Tets seem likewise to have a non-catalytic action on RNA. In favor of this view, we have recently reported that a catalytic activity-independent function of Tet2 is involved in regulating some retroviruses[30]. Specifically, we have shown in mouse ESCs that endogenous retrovirus (ERV) transcripts are regulated by two mechanisms: (1) post-transcriptional silencing of ERV RNAs via Tet2-mediated RNA hydroxymethylation and (2) transcriptional repression of ERVs through binding of Tet2 to RNA and concomitant recruitment of HDAC activity. Understanding the genomic characteristics that distinguish Tet1/2-bound sites that do not have 5hmC will require further study. Our first analyses suggest that at least some Tet1/Tet2-bound RNAs that do not have 5hmC display distinct DNA methylation patterns within the gene bodies of the corresponding loci.

Our study uncovers an unrecognized role of Tet-mediated RNA hydroxymethylation as a mark contributing, through mRNA destabilization, to the transcriptome flexibility required for embryonic stem cell differentiation. This role appears to be opposite to that reported for 5mC, the 5hmC precursor. Among the recently reported effects of m5C on mRNA fate[47–49] (e.g., mRNA nuclear export, viral RNA splicing and translation), it has been shown in both physiological and pathological contexts that m5C enhances mRNA stability[50,51]. This opposite role of 5hmC as compared to its precursor suggests that RNA hydroxymethylation is an important post-transcriptional modification with specific functions affecting mRNA metabolism. Accordingly, we show here that Tet-mediated hydroxymethylation can lead to downregulation and upregulation, destabilization, and splicing of modified transcripts. Considering the major roles of writers and readers in determining the regulatory roles of RNA modifications, it will be interesting in the future to characterize 5hmC effectors, in order to better understand the context-dependent functions of this mark, as for m6A[52]. Besides affecting stability, it seems that 5hmC might also impact RNA splicing. First, we found both chromatin-associated and intronic regions of presumably unspliced nascent pre-mRNAs to be rich in 5hmC. This suggests, as we have reported previously[30], that 5hmC deposition might occur co-transcriptionally. Second, we found Tet-deposited 5hmC to correlate with a higher ratio of spliced to unspliced transcripts. While Tet-deposited 5hmC could have a role in splicing per se, this might also partly explain the impact of 5hmC on stability. In support of this, it has been reported, for example, that the half-life of the intron-less chemokine *CXCL1* mRNA is shorter than that of the corresponding intron-containing control[53]. We propose a role of 5hmC as an intronic pre-mRNA modification promoting splicing and leading to a fast turnover of transcripts. The above hypothesis deserves future study.

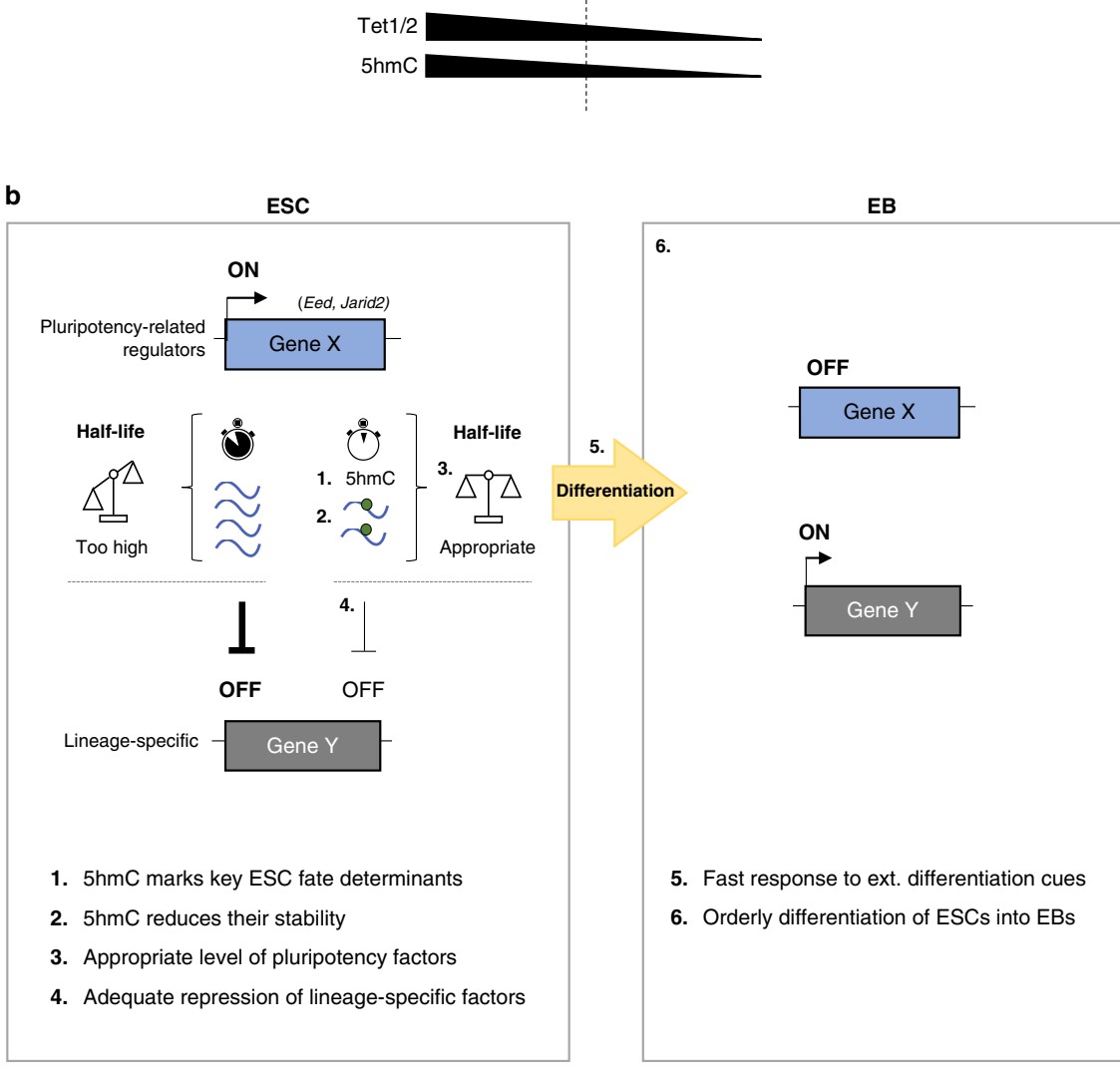

**Fig. 7 Model of Tet-mediated 5hmC as a mark of transcriptome flexibility in ESCs. a** Scheme illustrating that spontaneous ESC-to-EB differentiation leads to reduced *Tet1* and *Tet2* expression and to a marked decrease in 5hmC. **b** 5hmC mRNA modification acts as an essential regulatory layer to safeguard efficient, timely, authentic downregulation of lineage-specific genes. 1. In WT ESCs, Tet1 and Tet2 hydroxymethylate many RNA transcripts, such as those encoding key pluripotency-related regulators (e.g. *Eed, Jarid2*). 2. 5hmC results in transcript destabilization. 3. This leads to an appropriate expression level (not too high, not too low) of these key cell fate determinants. 4. Consequently, this would ensure adequate repression of lineage-specific factors. 5. This controlled repression critically prepares ESCs to rapidly respond to external cues. 6. This stepwise model would ensure orderly differentiation of ESCs to EBs.

In conclusion, our study uncovers an unrecognized role of Tet-mediated RNA hydroxymethylation as a mark contributing to the transcriptome flexibility required for embryonic stem cell differentiation. In other words, our work reveals 5hmC as a timely maintainer of the balance between pluripotency and lineage-priming factors, thus ensuring orderly differentiation of ESCs. Post-transcriptional RNA modifications such as m6A[31,54,55] and 5hmC should be regarded as constituting a crucial layer involved in fine-tuning gene expression in order to regulate stem cell function and developmental processes.

## Methods

**Cell culture**. Mouse ESCs[20,21,37] were grown under standard culture conditions. Briefly, cells were cultured on 0.1%-gelatin-coated tissue culture plates in high-glucose DMEM-containing 15% FBS, 1 mM sodium pyruvate, 1% non-essential amino acids, 1% glutaMAX™, 100 U/ml penicillin, 100 μg/ml streptomycin, 0.1 mM

β-mercaptoethanol, and 1000 units/ml recombinant mouse leukemia inhibitory factor (LIF) (All reagents from Gibco, except LIF from Millipore).

**Embryoid body formation**. Embryoid bodies (EBs) were obtained by spontaneous differentiation of ESCs[36]. Briefly, ESCs were trypsinized, resuspended in ES medium without LIF, and counted with a TC20™ Automated Cell Counter (BIO-RAD). Then $4 \times 10^6$ ESCs were seeded onto Greiner Petri dishes (Greiner) in 15 ml ES medium without LIF. EBs were maintained in ES medium without LIF for four days before collection for further analysis.

**CRISPR-Cas9 tagged Tet1, Tet2, and Tet2ΔRBD ESCs**. Mouse ESCs producing tagged Tets were generated with the CRISPR-Cas9 nuclease system[56]. Briefly, sgRNAs were designed to target the stop codons of *Tet1* and *Tet2* (for C-terminal tags) using the guidelines described in MIT's online tool (http://crispr.mit.edu). They were cloned into the pX461 vector. Lipofectamine™ 3000 was used according to the manufacturer's instructions (ThermoFisher Scientific) to co-transfect ESCs with a sgRNA-containing plasmid and a template containing triple tags (Flag + HA + Twin-Strep) from the pINTO-N3 vector[38], flanked by homologous arms for *Tet1* and by homologous arms with or without RBD for *Tet2*. 24 h after

transfection, individual ESCs were seeded into 96-well plates via serial dilution. One week later, clones were picked and analyzed for the Flag-tag by western blotting, and the CRISPR-Cas9-targeted genomic regions were PCR-amplified and sequenced in clones producing tagged Tet1, Tet2, or Tet2ΔRBD. All relevant sgRNA sequences and primers are listed in Supplementary Data 8.

**Cell fractionation**. ESCs were washed twice with cold PBS. The cell pellet was lysed with Igepal lysis buffer (10 mM Tris pH 7.4, 150 mM NaCl, 0.15% Igepal CA-630) and incubated on ice for 5 min. The lysate was then gently overlaid on top of sucrose buffer (10 mM Tris pH 7.4, 150 mM NaCl, 24% sucrose). After centrifugation at $3500 \times g$ for 10 min at 4 °C, the supernatant was saved for cytoplasmic RNA extraction. The pellet containing cell nuclei was briefly rinsed with cold PBS-EDTA (0.5 mM) and resuspended in glycerol buffer (20 mM Tris pH 7.4, 75 mM NaCl, 0.5 mM EDTA, 50% glycerol). This was followed by the immediate addition of urea buffer (10 mM Tris pH 7.4, 300 mM NaCl, 7.5 mM MgCl$_2$, 1 M urea, 0.2 EDTA, 1% Igepal CA-630) and incubation on ice for 2 min. After centrifugation at $13,000 \times g$ for 2 min at 4 °C, the supernatant for nucleoplasmic RNA extraction was collected and the chromatin pellet was further processed with TURBO DNase followed by Proteinase K treatment before RNA extraction.

**RNA and DNA extraction**. Total RNA was extracted with the RNeasy Mini or RNeasy Maxi Kit (Qiagen) or with TRIzol (ThermoFisher) according to the manufacturer's instructions. Genomic DNA was eliminated by DNase I treatment. Isolated RNA was used for downstream quantitative PCR, mass spectrometry, and hMeRIP-Seq. Genomic DNA was extracted with the DNeasy Blood & Tissue Mini Kit (Qiagen).

**Reverse transcription coupled to quantitative PCR**. Isolated RNA was converted to cDNA with qSCRIPT (Quanta). Gene expression was analyzed with the LightCycler 480 SYBR Green I Master mix (Roche) on the LightCycler 480 real-time PCR system (Roche). In all cases, average threshold cycles were determined from at least duplicate reactions, and gene expression levels were normalized to those of a housekeeping gene as indicated (*18S* rRNA, or *Gapdh*). The primers used in this study are shown in Supplementary Data 8.

**Western blot analysis**. Cells were harvested by scraping and lysed with IPH buffer containing EDTA-free Protease Inhibitor Cocktail (Roche). Cell extracts were fractionated by SDS-PAGE and transferred to PVDF membranes for immunostaining. Membranes containing the transferred proteins were blocked with 5% (w/v) non-fat dried skimmed milk powder (Bio-rad) in PBST and then incubated overnight at 4 °C with primary antibody against Flag-tag (1:2000, Sigma #F1804) in blocking buffer. The membranes were washed three times with PBST for 10 min and incubated with a 1:10,000 dilution of horseradish-peroxidase-conjugated anti-mouse or anti-rabbit antibodies for 1 h. They were then washed with PBST three times and developed with the ECL system (Amersham Biosciences) according to the manufacturer's protocols. Original images for all western blots are supplied as Source Data File.

**Dot blotting for 5hmC quantification**. RNA and DNA were extracted and spotted onto a nylon membrane (GE Healthcare Hybond-N+). The membrane was dried and cross-linking was performed twice with 200,000 μJ/cm$^2$ UV. For quantification, the membrane was stained with 0.04% methylene blue in 0.5 M sodium acetate and rinsed with PBS + 0.1% Tween-20 for 5 min. It was then blocked in 3% (w/v) non-fat dry milk in PBS + 0.1% Tween-20 for 1 h, transferred into a blocking solution supplemented with rat anti-5hmC monoclonal antibody (Diagenode #MAb-633HMC) diluted 1:500 and incubated overnight at 4 °C. Thereafter, the membrane was washed three times with PBS + 0.1% Tween-20 for a total of 30 min. It was transferred into a blocking solution supplemented with HRP-linked anti-rat IgG (Abcam #Ab6734) diluted 1:1000, incubated for 1 h at room temperature, washed three times with PBS + 0.1% Tween-20, and developed with the ECL system (Amersham Biosciences) according to the manufacturer's protocols. ImageJ software was used for signal quantification. Original images for all dot blots are supplied as Source Data File.

**LC–MS/MS for 5mC and 5hmC detection and quantification**. Mass spectrometry analysis was performed as described previously[26]. Briefly, 3 μl of 10× buffer (500 mM Tris-HCl, 100 mM NaCl, 10 mM MgCl$_2$, 10 mM ZnSO$_4$, pH 7.0), 2 μl (180 units) of S1 nuclease, 2 μl (0.001 units) of venom phosphodiesterase I, and 1 μl (30 units) of CAIP were added to 10 μg of total RNA from WT ESCs and TKO ESCs (in 22 μl of H$_2$O). The mixture (30 μl) was incubated at 37 °C for 4 h. The resulting solution was extracted with chloroform three times. The upper aqueous phase was collected and passed through a solid-phase extraction cartridge filled with 50 mg of sorbent of graphitized carbon black to remove the salts. The elution was then dried with nitrogen gas at 37 °C for subsequent chemical labeling and LC–ESI-MS/MS analysis by an AB 3200 QTRAP mass spectrometer (Applied Biosystems, Foster City, CA, USA).

**Vitamin C, H$_2$O$_2$, and BSO treatments**. ESCs in culture were treated with 50 μM Vitamin C (Sigma) for 16 h[4]. TKO ESCs were treated with 20 μM hydrogen peroxide[57] (H$_2$O$_2$, Sigma) for 24 h or with 500 μM buthionine sulfoximine[58] (BSO, Sigma) for 48 h. In each experiment, an equal volume of vehicle (water) was used as a control. Cells were collected after washing with PBS and processed for dot blotting.

**In vitro transcription**. In vitro transcription was performed with the MEGAscript® T7 Transcription Kits (Life Technologies) according to the manufacturer's instructions. For methylated and hydroxymethylated transcripts, ribo-CTP nucleotides were replaced in the reaction with ribo-5mCTP or ribo-5hmCTP (TriLink Biotechnologies). The DNA fragment containing TC-rich motifs was synthesized by IDT and subsequently cloned into a plasmid containing a T7 promoter. The integrity of the IVT-produced transcripts was confirmed with an Agilent 2100 Bioanalyzer and these transcripts were used later in antibody validation in hMeRIP-qPCR and in vitro stability assay.

**Hydroxymethylated RNA immunoprecipitation (hMeRIP)**. The procedure was performed on ESCs (WT, TKO, tagged Tet2WT, and Tet2ΔRBD) and EBs (WT) as described previously[28]. Briefly, 1 mg total RNA was fragmented to an average size of 200–300 bp. It was then precipitated in ethanol, resuspended in RNase-free ddH$_2$O, and the fragmentation efficiency was checked on a Bioanalyzer RNA chip (Agilent). For immunoprecipitation, RNA fragments only or fragments spiked with 2.5 μg IVT-produced transcripts containing UC-rich motifs with distinct RNA modifications (C, 5mC, and 5hmC), were denatured by heating at 70 °C for 5 min, chilled on ice for 5 min, and then incubated overnight at 4 °C with or without 12.5 μg anti-5hmC antibody (Diagenode monoclonal #MAb-633HMC) in freshly prepared 1X IP buffer (50 mM Tris-HCl pH = 7.4, 750 mM NaCl and 0.5% Igepal CA-630, RNasin 400 U/ml and RVC 2 mM) supplemented with protease inhibitors (cOmplete, Mini, EDTA-free, Roche). Samples were then incubated at 4 °C for 2.5 h with 60 μl equilibrated Dynabeads Protein G (Life Technologies), washed three times for 5 min with 1 ml IP buffer, and eluted by addition of 1 ml TriPure Reagent (Roche). This was followed by RNA extraction according to the manufacturer's instructions. Samples were then subjected to deep sequencing and the spike-ins were analyzed by quantitative PCR (primers available in Supplementary Data 8). All hMeRIP-Seq and qPCR experiments were performed in triplicate.

**RNA immunoprecipitation**. RNA immunoprecipitation (RIP) was performed using Magna RIP$^{TM}$ RNA-Binding Protein Immunoprecipitation Kit (Millipore) following the manufacturer's instructions. Briefly, cytoplasmic extract from ~1 × 10$^7$ tagged ESCs was distributed equally among samples and controls. For sample reactions, 10 μg of anti-flag antibody (Sigma, #F1804) was used for 75 μl of magnetic protein G beads. For control reactions, 10 μg of mouse IgG (Millipore, #12-371) with no immunoreactivity was used for 75 μl of magnetic protein G beads. After stringent washes and proteinase K digestion, immunoprecipitated RBP/RNAs (RIP) and total RNA (Input) from ESCs were subjected to downstream library preparation. All RIP-Seq experiments were performed at least in duplicate.

**Library preparation and deep sequencing**. 5 to 10 ng dsDNA was subjected to 5′ and 3′ protruding end repair, followed by the addition of non-templated adenines to the 3′ ends of the blunted DNA fragments, allowing ligation of Illumina multiplex adapters. The DNA fragments were then size-selected so as to remove all unligated adapters and to sequence 200–300-bp fragments. Eighteen PCR cycles were carried out to amplify the library. DNA was quantified by fluorometry with Qubit 2.0 and DNA integrity was assessed with a 2100 Bioanalyzer (Agilent). Six picomoles of the DNA library spiked with 1% PhiX viral DNA were clustered on cBot (Illumina) and then sequenced on a NextSeq500 (Illumina).

**Preprocessing of sequencing data**. Unless specified differently, sequencing data were preprocessed using the following steps: the raw sequencing data were first analyzed with FastQC (Andrews, 2010, https://www.bioinformatics.babraham.ac.uk/projects/fastqc/). Low-complexity reads were removed with the AfterQC tool[59] with default parameters. To get rid of reads originating from rRNA or tRNA, the reads were mapped to mouse tRNA and rRNA sequences with Bowtie2[60]. The rRNA and tRNA sequences were downloaded from https://www.ncbi.nlm.nih.gov/nuccore using *Mus musculus* [organism] AND (biomol_rrna [PROP] OR biomol_trna [PROP]) as search parameters. Reads that did not map to tRNA or rRNA sequences were then further processed with Trimmomatic[61] using default parameters to remove adapter sequences. The resulting fastq data were again analyzed with FastQC to ensure that no further processing was needed.

**hMeRIP-sequencing analysis**. Raw mouse ESCs and EBs hMeRIP-sequencing reads were preprocessed as described in the previous section. Pre-processed reads were then mapped against the mouse reference genome (mm9) with the STAR algorithm[62] using the RefSeq reference transcriptome (downloaded on March 2012). 5hmC peak regions were identified by applying the MACS2 peak-calling tool[63] onto immunoprecipitated (IP) samples, using their input counterpart to

estimate background noise (*q*-value < 0.05). It is worth noting that the "expected genome size" MACS2 parameter was set as the sum of all transcript lengths, including both exons and introns (counting regions shared by several transcripts only once), and summit positions were identified using the MACS2 "-call-sum-mits" option. To avoid identifying extremely large peak regions, the peaks were resized to 100 bp on both sides of the identified summit. So-called "expected peaks" (regions with a high read count and therefore most likely to generate peaks) were also generated by applying MACS2 with the same parameters to the input only (using MACS2 background modeling). A "bedtools intersect"-based in-house script was then used to identify 5hmC-modified regions observed in all replicate experiments[64]. These replicated peaks were reported as the final list of "5hmC peaks" (Supplementary Data 1) (replicated "expected peaks" were also generated by the same approach). Finally, a metasample combining the mapped reads of all the replicates was generated for each condition. To obtain visual representations of local enrichment profiles, bedgraph files were generated from mapped metasample files (bam) and uploaded into the IGV tool[65]. For differential analysis, reads from metasamples were counted in each "replicated peak" using the FeatureCounts algorithm on IP and input samples from each condition and normalized as reads per kilobase per million (RPKM). Enrichment ratios were defined for each condition as IP over input RPKM levels. Peaks were reported as differentially marked if a fold change of at least 1.5 was observed between the enrichment ratios of the two conditions.

**Motif analysis of hMeRIP peaks.** To perform the motif analysis, 5hmC and expected peaks were associated with transcripts with "bedtools intersect"[64] on the RefSeq transcriptome. The strand of each peak was attributed to its associated transcript (unassociated peaks were ignored and peaks intersecting transcripts of both strands were duplicated). Then the peaks were extended to 250 bp on both sides of the center and "bedtools getfasta"[64] was used to extract peak sequences in a stranded way. The meme-suite[66] (http://meme-suite.org) was then used for motif analysis. A first-order Markov model was generated using the "fasta-get-markov" function on the sequences from the input sample. Then the "meme" tool was used to identify top overrepresented motifs, using the aforementioned Markov model as a background model and expected peaks as negative control peaks. The number of motifs was restricted to 10 and the MEME search window was set between 5 and 12. Finally, we used "Centrimo" to evaluate the position of the motif relatively to the peak center, and decentered motifs were excluded.

**Distribution of hMeRIP peaks.** The "5hmC" and "expected" peaks identified by hMeRIP-Seq were annotated with the RefSeq gene annotation. Peaks were assigned to one or several transcripts and to annotated structural elements: to an exon when the peak summit was inside an annotated exon, to an intron when the peak summit was outside the exon but inside the transcript. Peaks that could not be associated with a coding gene or that could not be uniquely associated with one of these categories (e.g., ambiguous annotation due to overlapping transcripts) were left unannotated. The same rules were used to categorize peaks according to their association with coding sequences (CDS) or flanking regions (5′UTR and 3′UTR). For each transcriptomic region, the enrichment in 5hmC peaks was evaluated as the difference between the observed and expected percentages of 5hmC peaks in that region.

**RIP sequencing analysis.** Raw reads were processed as described in the "Pre-processing of sequencing data" section of this manuscript. The processed data were then mapped to the mouse genome (mm9), using the RefSeq reference tran-scriptome (downloaded on March 2012) and the RSEM tool[67]. Transcripts Per Million normalized counts (TPM) were computed from the RSEM expression counts and a pseudocount of 1 TPM was added and a transcript with higher TPM value in IP over Input was considered as Tet-enriched.

**Comparison of Tet1/2-bound targets (using published data).** MeDIP-Seq[19] were downloaded as raw data from the SRA database (https://www.ncbi.nlm.nih.gov/sra) (ERP000570).

Raw data were preprocessed as described under "Preprocessing of sequencing data" (without the rtRNA filtering step). MeDIP data were filtered for duplicate reads by means of the picard tool MarkDuplicates (http://broadinstitute.github.io/picard/) and mapped with bowtie2[60]. Peaks were identified with the MACS2 peak-calling tool[63] (*q*-value < 0.05; expected genome size set as 'mm') and summit positions were identified with the "–call-summits" MACS2 option. Annotation was finally done with a bedtools-based script (the corresponding region was counted as a gene if the peak fell between a TSS and a TTS).

Transcripts bound to Tet1 and/or 2 were intersected with the 5hmC-containing transcripts identified in hMeRIP experiments to define 5hmC-modified and unmodified Tet1/2-bound transcript categories. For each MeDIP sample (*n* = 2), the transcripts identified using the aforementioned annotation process were intersected with each of the two categories and the percentages of 5hmC-marked and unmarked Tet-bound transcripts were computed. A *t*-test was then applied to compare the percentages obtained for each category.

**mRNA stability and translation efficiency analyses.** In order to restrict the mRNA stability and translation efficiency analyses to expressed genes, we evaluated gene expression in wild-type mouse ESC cells. For this, the Poly-A RNA-Seq data were first preprocessed as described in the "Preprocessing of sequencing data" section and mapped to the mm9 genome using STAR tool[62] with the RefSeq transcriptome. Then gene expression was computed with the HTseq tool[68] and converted to TPM. Genes showing more than 1 TPM were considered expressed. We then stratified the transcripts of expressed genes into 5hmC-marked and unmarked on the basis of the presence, within the transcript, of at least one 5hmC peak from the hMeRIP-Seq analysis. Finally, external mRNA stability microarray data[40] and ribosome-sequencing profiles[39] in wild-type mouse ESCs were used to compare the mRNA half-lives and translation efficiencies of 5hmC-modified and non-modified transcripts with a Wilcoxon test.

**RNA transfection.** For the in vitro stability assay, unmodified and 5hmC-modified IVT transcripts were delivered into WT ESCs with the JetPrime polyplus reagent (Polyplus transfection) according to the manufacturer's instructions. This was followed by quantitative PCR analysis at 6 and 24 h post-transfection[69].

**α-Amanitin treatment.** For the in vivo stability assay, α-amanitin treatment of ESCs was performed. Briefly, WT, TKO, Tet2WT, and Tet2ΔRBD ESCs were treated with 10 μg/ml α-amanitin (Santa Cruz) or with an equal volume of vehicle (water) as a control for 0 or 4 h, respectively. For the rescue experiments, TKO ESCs were transfected with the Tet2FL or Tet2 catalytic mutant (Tet2Mut)[30] plasmid with JetPrime polyplus reagent according to the manufacturer's instructions. They were then treated with α-amanitin as described above. The cells were then collected after washing with PBS and processed for quantitative PCR analysis and/or RNA-Seq. For RNA-Seq, total RNA was extracted from α-amanitin-treated cells and untreated control cells and depleted of ribosomal RNA. The RNA in this fraction was fragmented before library preparation and deep sequencing, as described above. All primers used in this study are described in Supplementary Data 8.

**RNA-Seq analyses for differential expression and splicing.** Sequencing reads were preprocessed as described under 'Preprocessing of sequencing data'. Pre-processed reads were then mapped against the mouse reference genome (mm9) with the STAR algorithm[62] using the RefSeq reference transcriptome (downloaded on March 2012). Then gene expression was computed with the HTseq tool[68]. Raw gene expression counts were then subjected to DESeq2[70] for normalization and analysis of differential expression analysis between control (WT and TKO ESCs) and α-amanitin-treated cells. Similar conditions were used for splicing. IR Finder version 1.2.3[71] was applied to detect unspliced and spliced transcripts. Count data from processed bam files were obtained with featureCounts[72] and then converted to FPKM. Genes with FPKM > 0 were considered expressed. Only expressed genes containing intronic 5hmC peaks were selected and further overlapped with IR Finder output. The ratio of unspliced to spliced reads from the intersection was quantified with Bedtools[64]. The data were normalized to the unspliced/spliced ratio found for untreated cells at time 0 h.

**Statistics and reproducibility.** Statistical analysis was performed using either the computing environment R or GraphPad Prism 7. Unless otherwise indicated, all experiments included technical replicates and were repeated at least three inde-pendent times. All statistics were evaluated by Student's *t*-test unless specified otherwise. Data and graphs are presented as means ± SEM. The statistical sig-nificance criterion was *P* < 0.05.

**Reporting summary.** Further information on research design is available in the Nature Research Reporting Summary linked to this article.

## Data availability

The RNA-Seq, hMeRIP-Seq, and RIP-Seq data supporting the findings of this study have been deposited in the GEO repository under the accession code "GSE131902". The stemness/pluripotency signature genes were derived from the ESCAPE[34] and the StemChecker[33] databases and from published data[31,32,35]. The microarray data[40] "Supplementary Table 1 [https://doi.org/10.1093/dnares/dsn030]", Ribo-Seq[39] "Supplementary Table S1C [https://doi.org/10.1016/j.cell.2011.10.002]", and MeDIP-Seq[19] "ERP000570" supporting our study are published data. Source data are provided with this paper.

## Code availability

Code supporting this study is available at a dedicated Github repository [https://github.com/martinBizet/hmC_ES].

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

## Acknowledgements

We thank Prof. Rudolf Jaenisch (Cambridge, USA) for the mouse wild type, *Tet1/2* double knockout (DKO), and *Tet1/2/3* triple knockout (TKO) ESCs, Prof. Anjana Rao for *Tet3* knockout mESCs, and Prof. Dr. Roberto Bonasio for the pINTO-N3 vector. We also thank Mathieu Defrance and Romy Chen-Min-Tao for scripts development. J.L. was supported by BELSPO and by the Belgian "Fonds de la Recherche Scientifique" (FNRS). N.R., A.P., E.B., and B.H. were supported by the FNRS, and A.L.G. by the "Télévie". N.K. S. was supported by the ULB Foundation. F.F. is a ULB Professor. F.F.'s lab was funded by grants from the FNRS and Télévie, the "Action de Recherche Concertée" (ARC) (AUWB-2018-2023 ULB-No 7), Wallon Region grants U-CAN-REST and INTREPID (1710179-WALLINOV, INTREPID RW 7787), an FNRS Welbio grant (FNRS-WELBIO-CR-2017A-04 and FNRS-WELBIO-CR-2019A-04R), the ULB Foundation and the Belgian Foundation against cancer (FCC 2016-086 FAF-F/2016/872). P.J.H. is supported by the University of Chicago Medical Scientist Training Program (MSTP; NIH MSTP training grant T32GM007281). S.N. is an HHMI fellow of the Damon Runyon Cancer Research Foundation (DRG-2215-15). Research from the M.F. laboratory was supported by Spanish Agencia Estatal de Investigación co-funded by the FEDER Program of EU (BFU2016-80899-P)(AEI/FEDER, UE) and Ramón y Cajal award (RYC-2014-16779). J.J.L.W. holds a Fellowship from the Cancer Institute of New South Wales, Australia. J.J.L.W.'s lab was supported by grants from the NHMRC Australia (1128175 and 1129901) and the Cancer Council of NSW Australia (RG19-05). J.W.'s lab was funded by grants from NYSTEM (C32569GG; C32583GG) and NIH (R01GM129157; R01HD095938; R01HL146664).

## Author contributions

J.L., N.R., and F.F. designed the experiments and interpreted the data. J.L., N.R., R.D., F.M., B.H., and P.P. generated tagged Tet1, Tet2, and Tet2ΔRBD ESC lines and performed cellular RNA fractionation, in vitro transcription, BSO and $H_2O_2$ treatments, RIP, western blots, RT-qPCR, and dot blots. S.N. and P.J.H. helped with RIP-Seq experiments. M.B., N.K.S., G.M., and R.S. performed bioinformatics analyses. E.C. performed antibody validation, hMeRIP, and deep-sequencing experiments. E.B. and A.L.G performed vitamin C treatment and related dot blots. D.G., A.F.-I., J.W., and M.F. performed α-amanitin treatments, RNA stability assays, and rescue experiments. C.M. and B.Y. performed mass spectrometry analyses. J.J.L.W. performed paired-end RNA-Seq. J.L., N.R., and A.P. prepared the figures and assembled the revised manuscript. F.F. wrote the manuscript.

## Competing interests

The authors declare no competing interests.
