## [Peer Review File · Nature Communications]

Reviewers' comments:

Reviewer #1 (Remarks to the Author):

In this study Fuks and colleagues report that Tet enzymes hydroxylate pluripotency associated mRNAs, which reduces their stability and is relevant to ESC differentiation. While RNA hydroxylation and role of Tet enzymes in this process was previously established, the novelty of this study is in identifying which classes of transcripts are hydroxylated, and what effects hydroxylation has on their stability and ESC differentiation. The study is very interesting and I support its publication. However, there are several questions that need to be clarified, a couple of experiments need to be performed, and finally conclusions need to be drawn more moderately in the light of the data available. The authors do not provide a lot of differentiation or rescue experiments, so it is better to keep the conclusions moderate and some rewording will help.

Major comments:

1- A lot of 5hmC is seen in intronic regions followed by coding regions. What is the significance of 5hmC at intronic regions? Does that also correlate with pluripotency transcripts? The authors can elaborate more on this with some further data analysis and comments in the discussion.

2- The total Tet1 bound and Tet2 bound mRNA sites are about 15K, while there are only 1600 5hmC marked sites in a total of ~700 transcripts. Why are Tet1 and Tet2 binding on sites that are not hydroxylated? what are they doing at those sites? Did the hMeDIP really capture all the hydroxylated regions? Authors can comment and clarify this for reviewers and readers.

3- Only 64% of 5hmC sites overlap with Tet1/2 binding sites, how did the other 36% get hydroxylated? Elaboration is needed in line with point 3 raised above. Perhaps other enzymes or chemical processes involved here?

4- If we compare figure 3c-d, we find that comparison of WT vs dRBD Tet2 leads to a reduction of about 2000 mRNA targets, which is about 1/3rd. Why in figure 3e it is shown to be 47%? Even if it is 47%, the RBD seems to be partially critical for targeting Tet2 to RNA and its hydroxylation? Perhaps one way to test for this would be to rescue TKO ESCs RNA hydroxylation with Tet2 wild type and Tet2-dRBD to see how many sites are restored.

5- Was 5hmC seen only in down regulated transcripts? Any comments on if it was seen in up regulated transcripts too? Were Tet1/2 bound to any up regulated transcripts? Authors can do some more data analysis on this and comment in discussion.

6- Can the aberrant pluripotency transcript hydroxylation, stability and gene expression in TKO ESCs be rescued by Tet1 or Tet2, their CD, and their dRBD versions? Authors can perform some rescue experiments by overexpressing some or all of these Tet variants in TKO cells. Along the same lines, Can some in vitro RNA hydroxylation experiments be performed and stability of hydroxylated and unhydroxylated RNA be quantified in vitro?

7- Finally with the model: If the role of Tet1/2 is to hydroxylate pluripotency associated mRNAs to make them less stable: (1) Why is Tet1/2 and pluripotency mRNA hydroxylation so abundant in ESC state where pluripotency factor expression is high and pluripotency is maintained? (2) Likewise, the levels of Tet1/2 go down upon differentiation, how does this correlate with increased hydroxylation of

pluripotency transcripts during differentiation as reported by this study? Both are counter intuitive to the model proposed in the study? Major explaining is needed here, or alternative explanations perhaps.

Minor comments:

1-Is dRBD generated by deleting the whole RBD domain or only the 9aa? Please explain this clearly in the text.

2-Why is Tet3 not induced upon differentiation in the panels shown? It is known that Tet3 levels go up upon differentiation. Actually induction of Tet3 upon differentiation may serve as an explanation for major point 7-(2).

3- 5mC vs. 5hmC in RNA: If 5mC is the substrate for DNA hydroxylation, presumably in RNA it is also the substrate. What is the significance of RNA 5C methylation vs hydroxylation? RNA stability? Does loss of TET enzymes affect overall RNA methylation? Any comments and/or data will make the study more comprehensive.

Anonymous (I do not sign this review with my name)

Reviewer #2 (Remarks to the Author):

In this study, Fuks and colleagues have employed hydroxymethylated RNA immunoprecipitation followed by sequencing (hMeRIP-seq) to profile the distribution of 5hmC-enriched sites in the transcriptomes of wild-type, Tet1-3 TKO mouse embryonic stem cells (mESCs) and differentiating embryonic bodies (EBs). In addition, the authors used RIP-Seq to identify Tet1- and Tet2-associated mRNA transcripts and revealed that a previously reported RNA binding domain (RBD) of Tet2 is in part required for binding of RNA targets. RNA stability assays further suggest that 5hmC may promote RNA degradation of mRNAs encoding pluripotency-related factors in undifferentiated mESCs.

Overall, the results from this work confirm the presence of low levels of RNA hydroxymethylation in mammalian cells and tissues as previously reported (PMID: 25073028). While understanding the potential roles of Tet enzymes in RNA modifications and post-transcriptional regulation is of interest, this study, in its current form, did not provide sufficient conceptual advance, mechanistic insights, or methodological innovation required for publication at Nature Communication.

Specific comments:

1. Fig. 1C: first, the authors need to show the observed percentage of 5hmC at each category of genomic regions next to the expected percentage. Second, ~70% of 5hmC sites were identified to be within introns, which suggests that 5hmC is more enriched in nascent, un-spliced mRNAs. This is a consequential observation, and the authors should substantiate this by performing hMeRIP-seq with nascent or pre-existing pools of mRNAs using a metabolic labeling strategy.

2. Fig. 1b: Is the ctrl sample derived from input DNA (without IP) or mock IP using IgG? As antibodies for DNA modifications such as 5hmC have been shown to have intrinsic affinity for short unmodified repeat sequences (e.g. GTGTGTGTGT; PMID: 29941872), it is unclear whether the putative UC-rich motif is specifically associated with 5hmC sites or related to a 5hmC antibody specific artifact in hMeRIP-seq. It will be critical for the authors to use synthetic RNAs containing the UC-rich motif with distinct RNA modification states (unmodified, 5mC and 5hmC) to confirm specificity of 5hmC antibody in the hMeRIP-seq assay.

3. Supplementary Fig. 1d, statistical analysis of overlap between 5hmC transcripts and pluripotency-

related transcripts needs to be performed (e.g. hypergeometric test).

4. While catalytic domains of Tet1 can catalyze RNA hydroxymethylation in vitro, previous report indicates that only full-length Tet3 (but not Tet1/2) is enzymatically active in catalyzing 5hmC in RNAs in vitro (PMID: 25073028). Since Tet3 is expressed at very low level in mESCs, it is thus unclear whether 5hmC in mESCs is derived from low level Tet3 or low activity of Tet1/2. To dissect the underlying mechanism, the authors should use CRISPR knock-in strategy to introduce point mutations to inactive Tet1/2/3 individually or in combination to identify which Tet enzyme(s) is responsible for the 5hmC in mESC RNAs.

5. Fig. 2: Why are Tet enzymes only responsible for >50% of 5hmC in RNAs? The presence of appreciable levels of 5hmC in Tet TKO ESCs (>50% remaining) suggests that other enzymes(s) might be involved in oxidizing 5mC to 5hmC in mammalian cells. Some 5hmC may also be induced by cellular reactive oxygen species (ROS). It will be very helpful if the authors can experimentally test pharmacological means to manipulate ROS pathway and test whether this pathway contributes to 5hmC in RNA.

6. Fig. 3: Why the majority of Tet1- or Tet2-bound target RNAs do not contain 5hmC site? What is the difference between these two groups of Tet1/2 bound RNAs? The authors should further analyze the sequence context, expression level, RNA binding protein sites, other RNA modifications (e.g. 6mA) or other genomic features between these two groups of RNAs.

7. Fig. 4: In addition to four analyzed genes (Eed, Jarid2, Smarcc1, and Dab1), the authors should perform unbiased analysis of RNA stability of all transcripts (RNA-seq) after adding the transcription inhibitor.

Reviewer #3 (Remarks to the Author):

The authors revealed a novel function of Tet-mediated RNA 5hmC in the regulation of mouse ES differentiation. The authors made several interesting observations: 1) Hundreds of messenger RNAs were modified by 5hmC in ES cells; 2) Large number of transcripts showed a decreased 5hmC level during ES differentiation, partly in Tet enzymes dependence; 3) The mRNA targets bound by Tet1/Tet2 had similar topology like 5hmC sites: 4) Tet-mediated RNA 5hmC reduced mRNA stability. These discoveries are both novel and interesting. I would like to ask the following questions to be addressed in order to meet the standards for publication in this Journal.

1- The main novel part of this manuscript was the demonstration of 5hmC in promoting mRNA decay. Nevertheless, the same group has previously reported 5hmC regulates mRNA translation, and another Chinese group has showed that m5C stabilizes mRNA and regulates zebra fish early embryogenesis and human bladder carcinogenesis. The authors are encouraged to discuss the distinct roles of m5C and 5hmC in mRNA metabolism in the Discussion section.

2- The authors used Tet1/2/3 triple-knockout (TKO) mouse ESCs. Did the Tet1/2/3 family proteins have redundant mRNA targets? Which one of Tet proteins plays a major role in RNA m5C oxidation?

3- The authors may consider to apply 5hmC-targeted mRNA Reporter gene assays to provide supporting evidence for Tet-5hmC dependent hypothesis. If transfection is easy to be manipulated in ESC, this new piece of information will be a good addition to this work.

4- The out frame of figure panel can be deleted.

Reply to the referees:

Reviewer #1

We thank the reviewer for finding that “*the study is very interesting*» and that *he/she «supports its publication”*. We also appreciate his/her insightful suggestions. In the revised manuscript, we have addressed thoroughly all the comments of this reviewer. Notably, we have included many decisive controls, essential new results, and discussions as summarized below:

1. By performing loss-of-function experiments using CRISPR knock-in ESCs for Tet2 wt and Δ RBD, followed by hMeRIP-Seq, we confirm that hydroxymethylation of mRNAs is, at least in part, dependent on Tet2 RBD (Fig. 4 and Supplementary Table 6).
2. We have important new data confirming our initial experiments on the stability of key pluripotency transcripts. Using CRISPR knock-in ESCs for Tet2WT and Δ RBD or rescue of TKO ESCs with Tet2WT or a Tet2 catalytic mutant, we show that Tet2 RBD is required in part for transcript destabilization and that 5hmC deposition on these transcripts facilitates their degradation (Figs 6c and 6d and Supplementary Fig. 5).
3. We provide insights as to which Tets are responsible for 5hmC in RNAs in ESCs. Our novel experiments suggest that, while Tet3 seems not to be involved, both Tet1 and Tet2 appear to contribute similarly to RNA m5C oxidation and seem to have redundant functions (Fig. 3g and Supplementary Fig. 2a).
4. We describe our proposed model much more clearly and in much more detail. In brief, our data support a stepwise working model whereby 5hmC mRNA modification acts as an essential regulatory layer to safeguard efficient, timely, and authentic downregulation of lineage-specific genes (cf. Fig. 7 and Discussion).

Major comments:

1- «*A lot of 5hmC is seen in intronic regions followed by coding regions. What is the significance of 5hmC at intronic regions? Does that also correlate with pluripotency transcripts? The authors can elaborate more on this with some further data analysis and comments in the discussion* ».

We appreciate this comment regarding the significance of strong 5hmC enrichment within introns, which is also observed within pluripotency-related transcripts (Supplementary Fig. 1f). First, as our hMeRIP-Seq were performed on total RNA (which contains pre-mRNAs and mRNAs containing retained introns), we wondered whether 5hmC might be associated with nascent, unspliced pre-mRNAs. Hence, we assessed the level of 5hmC by dot blotting on the three following RNA fractions: nascent chromatin-associated, nucleoplasmic, and cytoplasmic. As shown in Supplementary Fig. 1g, we observed that chromatin-associated RNAs were readily hydroxymethylated, suggesting that 5hmC deposition may occur in a co-transcriptional manner, as we have previously suggested (Guallar et al. Nature Genetics 2018).

Second, we evaluated the role of Tet-deposited 5hmC in splicing regulation, by means of paired-end RNA-Seq in WT and TKO ESCs, followed by differential splicing analysis. We found Tet-deposited 5hmC to be associated with a higher ratio of spliced to unspliced transcripts (Supplementary Figs 1h and 1i and Supplementary Table 2). It is noteworthy that the 5hmC-enriched UC-rich motif we identified resembles motifs recognized by the pre-mRNA splicing regulator PTBP1 (ATtRACT database).

All in all, we believe that the involvement of Tet-deposited 5hmC in splicing could partly explain its observed impact on stability. In support of this, it has been reported, for example, that the half-life of the chemokine CXCL1 mRNA is reduced in the intron-less transcript as compared to its intron-containing control (Zhao and Hamilton JBC 2007). We propose a role of 5hmC as an intronic modification of pre-mRNA, promoting splicing and subsequently leading to fast turnover of transcripts. This hypothesis deserves future study.

We have now incorporated these novel results into the Main Text and Supplementary Data and have amended the Discussion.

2- «*The total Tet1-bound and Tet2 bound mRNA sites are about 15K, while there are only 1600 5hmC marked sites in a total of ~700 transcripts. Why are Tet1 and Tet2 binding on sites that are not hydroxylated? what are they doing at those sites? Did the hMeDIP really captured all the hydroxylated regions? Authors can comment and clarify this for reviewers and readers.*»

We appreciate these insightful comments. Regarding the questions «*why are Tet1 and Tet2 binding on sites that are not hydroxylated?*» and «*what are they doing at those sites?*», we believe that this suggests the interesting possibility that, besides hydroxylating mRNA, Tet1 and Tet2 might also function independently of their catalytic activity. This «*RNA hydroxymethylation independent*» role would be analogous to the well-described non-catalytic action of Tet1/Tet2 on DNA. For example, genome-wide DNA hydroxymethylation (hMeDIP) and ChIP-Seq have revealed that 5hmC-positive genes represent only 35% of Tet1-binding DNA targets (Williams et al. Nature 2011). On DNA, it is also well described that, in a catalytic-independent manner, Tet proteins associate with diverse chromatin-related machineries such as HDAC and

SET1/COMPASS, involved in transcriptional repression or activation (Delatte et al. EMBO J 2014).

Tets seem likewise to have a non-catalytic action on RNA. In favor of this view, we have recently reported that a catalytic-activity-independent function of Tet2 is involved in regulating some retroviruses (Guallar et al. Nature Genetics 2018). Specifically, we have shown in mouse ES cells that endogenous retrovirus (ERV) transcripts are regulated by two mechanisms: (1) post-transcriptional silencing of ERV RNAs via Tet2-mediated RNA hydroxymethylation; (2) transcriptional repression of ERVs through binding of Tet2 to RNA and concomitant recruitment of HDAC activity.

All in all, the above observations suggest that binding of Tet1/Tet2 to mRNAs that are not hydroxylated might provide a platform for epigenetic complexes such as HDAC, for transcriptional regulation. We now mention such catalytic-independent functions of Tet1/Tet2 on mRNAs, which invite future work, in the Discussion.

Concerning whether «*hMeRIP really captured all the hydroxylated regions*»: like any antibody-based immunoprecipitation NGS technology, there are some limitations as regards the capture of all modified regions. Hence, the number of RNA hydroxymethylated sites identified here is very likely an underestimate of the total number of such sites on mRNAs. Among several potential limitations for the detection of all 5hmC mRNA sites one can mention:

- RIP-Seq can be impacted by bioinformatic analysis (the choice of peak detection, alignment methods, etc... ; e.g. Helm and Motorin Nature Reviews Genetics 2017), and hence likely prevented detection of low-abundance 5hmC sites.
- The size of the RNA fragments used during the IP procedure can affect the binding of 5hmC antibody because this antibody requires more than just a single 5hmC for efficient binding. This is well known for hMeDIP-Seq, which uses the same antibody, and which has been shown to exhibit bias toward 5hmC-dense DNA regions (e.g. Skvortsova et al. Epigenetics & Chromatin 2017).
- The relative affinity of the antibody for 5hmC-modified RNA can affect the number of identified sites.
- Antibodies used in RIP-seq, such as ones against m6A (Schwartz et al. Cell 2013), could be intrinsically biased towards particular RNA sequences and secondary structures, so that other sites might not be identified.

3- « *Only 64% of 5hmC sites overlap with Tet1/2 binding sites, how did the other 36% get hydroxylated? Elaboration is needed in line with point 3 raised above. Perhaps other enzymes or chemical processes involved here?* »

This is a pertinent comment, as our data show indeed that Tet enzymes are only partly responsible for deposition of 5hmC in mRNA. This is clear both from the overlap between RIP-Seq for Tet1/Tet2 and hMeRIP-seq in WT ESCs (Fig. 3g) and from results of mass spec and hMRerIP-seq performed in Tet-knockout ESCs (Fig. 2). Our findings are consistent with earlier reports (Huber et al. ChemBioChem 2015, Fu et al. JACS 2014).

As asked, we have tested whether 5hmC might form on mRNA via other chemical processes. Specifically, we have evaluated whether 5hmC might be induced by cellular reactive oxygen species (ROS). This seems not to be the case, as treating ESCs with either buthionine sulfoximine (BSO) or H₂O₂ did not change global 5hmC levels in mRNA (Supplementary Fig. 2f). We cannot exclude that other chemical processes might be involved, and future work is warranted to test this possibility.

As also rightly suggested by this referee, 5hmC in RNA might be formed by enzymes other than Tets. Tets belong to the family of ferrous-ion- and α -KG -dependent dioxygenases (Fe²⁺ and 2-OG). This family includes more than 60 members, among which: (1) nucleic acid oxygenases (NAOXs), such as Tets and Alkbh enzymes (e.g. FTO); (2) hydroxylases; (3) JmjC-domain-containing enzymes, such as histone lysine demethylases. Interestingly, several of these enzymes are already known to modify various substrates (Klose et al. Nature Reviews Genetics 2006, Loenarz and Schofield Nature Chemical Biology 2008, Cloos et al. Genes Dev. 2008). Consequently, one way to start identifying novel RNA hydroxymethyltransferases might be: (1) to perform *in vitro* hydroxymethylation assays on TKO-cell extracts in the presence of various co-factors (e.g. Fe²⁺, α -KG, or FAD) and under different conditions (IDH1/2 mutations, presence of a 2-OG analogue or MAO inhibitor), and (2) to perform, in TKO cells, CRISPR KO screening targeting the whole above-mentioned family, with determination of relative 5hmC levels in RNA by mass spectrometry. Such work is clearly of interest, and should be done in the future.

We have amended the Main Text and the Discussion to reflect the above interesting points.

4- « *If we compare Fig. 3c-d, we find that comparison of WT vs dRBD Tet2 leads to a reduction of about 2000 mRNA targets, which is about 1/3rd. Why in Fig. 3e it is shown to be 47%? Even if it is 47%, the RBD seems to be partially critical for targeting Tet2 to RNA and its hydroxylation? Perhaps one way to test for this would be to rescue TKO ESCs RNA hydroxylation with Tet2 wild type and Tet2-dRBD to see how many sites are restored.* »

It seems that our initial description of these data was not clear and we apologise for this. In particular, the data in initial Fig. 3e were misleading, as they did not take into account all the Tet2-bound targets

observed upon deletion of RBD. We have now removed this panel and amended the Main Text to avoid confusion. This referee is thus right that Tet2 binding is about 30% reduced upon RBD removal (Figs 3c-e).

As asked, we have further tested our finding that the Tet2 RBD is partially required for RNA binding, targeting, and hydroxylation. We first attempted rescue experiments in TKO ESCs but, despite our efforts, this was a difficult task due to a low transfection efficiency. It was thus problematic to obtain enough transfected cells for hMeRIP-seq. Nevertheless, we performed loss-of-function experiments using CRISPR knock-in ESCs for Tet2 wt or Δ RBD, followed by hMeRIP-Seq. Interestingly, we observed a significant decrease (67.5%) in 5hmC-enriched regions upon deletion of Tet2 RBD (Fig. 4 and Supplementary Table 6). This confirms that hydroxymethylation of mRNAs is, at least in part, dependent on Tet2 RBD. Other mechanisms are likely at play to fully explain hydroxymethylation of mRNAs, such as a contribution of Tet1 or recruitment of Tet2 via other RNA-binding proteins, as we have previously described for retroviruses (Guallar et al. Nature Genetics 2018).

We have added these key and novel experiments in the Main Text.

5- « *Was 5hmC seen only in down regulated transcripts? Any comments on if it was seen in up regulated transcripts too? Were Tet1/2 bound to any up regulated transcripts? Authors can do some more data analysis on this and comment in discussion* ».

As asked, we have carried out further analyses. We performed RNA-Seq experiments in TKO ESCs and analysed upregulated or downregulated transcripts upon Tet depletion and made the following observations:

(i) By comparing 5hmC targets from hMeRIP-seq with Tet-regulated transcripts, we found 55.6% of the 5hmC-enriched targets to be upregulated and 44.4% to be downregulated (Supplementary Fig. 3f).

(ii) Comparison of Tet1/Tet2-bound mRNAs from RIP-Seq with RNA-Seq in TKO ESCs showed both up- (65.9%) and down-regulated transcripts (34.1%) (Supplementary Fig. 3g).

(iii) We also looked at the overlap between 5hmC-enriched targets bound by Tet1/Tet2 and up- or down-regulated transcripts. We observed that a significant number of downregulated transcripts harboring 5hmC were bound by Tet1/2 (68.2%). Many upregulated transcripts enriched in 5hmC also interacted with Tet1/Tet2 (67.3%)(Supplementary Fig. 3h).

These results suggest that Tet-mediated RNA hydroxymethylation can lead not only to downregulation, as we initially showed, but also to upregulation. Either a negative or a positive impact of an RNA modification according to the regulated transcript has also been described for m6A (Shi et al. Molecular Cell Review 2019). Our results also suggest co-transcriptional deposition of the 5hmC modification within RNA, as recently shown for m6A (Huang et al. Nature 2019, Liu et al. Science 2020). As requested, we have incorporated these novel results in the Main Text as well as in the supplementary data and have amended the Discussion.

6- « *Can the aberrant pluripotency transcript hydroxylation, stability and gene expression in TKO ESCs be rescued by Tet1 or Tet2, their CD, and their dRBD versions? Authors can perform some rescue experiments by overexpressing some or all of these Tet variants in TKO cells. Along the same lines, Can some in vitro RNA hydroxylation experiments be performed and stability of hydroxylated and unhydroxylated RNA be quantified in vitro?* »

As asked, we have performed rescue experiments on aberrant pluripotency transcripts. Specifically, we did transcription inhibition with α -amanitin to assess the stability of the key pluripotency transcripts initially studied: Eed, Jarid1, Dab1, and Sfpq. Rescue experiments on TKO ESCs were carried out with Tet2WT or a Tet2 catalytic Mutant (Tet2Mut). We found wild-type Tet2, but not Tet2Mut, to rescue the mechanism that destabilizes Eed, Jarid2, Dab1, and Sfpq in WT ESCs (Figs 6c and 6d and Supplementary Fig. 5a). Of note, we did not attempt similar rescue experiments to assess gene expression, as RNA-Seq in TKO ESCs revealed unchanged transcript-level expression of Eed, Jarid2, Dab1, and Sfpq (data not shown). These important novel results further suggest that 5hmC deposition on these transcripts facilitates their degradation.

For Tet2 Δ RBD, we made use of our CRISPR knock-in ESC model, for which we obtained novel hMeRIP-Seq data (cf. comment 4). We found that while Tet2WT decreased the abundance of pluripotency transcripts, this was no longer the case upon deletion of Tet2 RBD (Supplementary Fig. 5b). These results indicate that Tet2 RBD is required, at least in part, to destabilize key pluripotency transcripts in ESCs.

Concerning the question « *Can some in vitro RNA hydroxylation experiments be performed and stability of hydroxylated and unhydroxylated RNA be quantified in vitro?* »: we have produced unmodified and 5hmC-modified transcripts by *in vitro* transcription in the presence of C or 5hmC nucleotides and have used them to transfect WT ESCs. Their abundance was measured 6h and 24h post-transfection in order to evaluate their relative stability. These time points were chosen according to previously reported work and are described as appropriate for studying transcript stability in ESCs (Wroblewska et al. Nature Biotechnology 2015). We observed after ESC transfection that *in vitro* 5hmC-modified transcripts were less stable than their unmodified counterparts (Figs 6a and 6b). These *in vitro* data are in good agreement with our original finding that 5hmC favors fast turnover of RNA transcripts.

We have now introduced these new results as a brand new Main Figure (Fig. 6) and have amended the Main Text.

7- « Finally with the model: If the role of Tet1/2 is to hydroxylate pluripotency associated mRNAs to make them less stable: (1) Why is Tet1/2 and pluripotency mRNA hydroxylation so abundant in ESC state where pluripotency factor expression is high and pluripotency is maintained? (2) Likewise, the levels of Tet1/2 go down upon differentiation, how does this correlate with increased hydroxylation of pluripotency transcripts during differentiation as reported by this study? Both are counter intuitive to the model proposed in the study? Major explaining is needed here, or alternative explanations perhaps. »

First, this referee might have missed some of our initial data, as we did show that ESC-to-EB differentiation leads, concomitantly with reduced Tet1 and Tet2 expression, to a marked decrease, rather than increase, in 5hmC (original Fig. 1f-i). To avoid confusion, we have clarified the text accordingly. In brief, we initially showed by dot blotting that EBs displayed a lower global 5hmC signal than ESCs (new Fig. 1h and Supplementary Fig. 1k). Likewise, we observed by hMeRIP-Seq that 5hmC marking was decreased in over 80% of the transcripts upon ESC-to-EB differentiation (Figs 1i and 1j and Supplementary Figs 1l and 1m).

Regarding our model, its description was apparently unclear, and we apologize for this. Below we describe the proposed model more clearly and in more detail, and this new description is now incorporated into the Discussion. We have also modified the model in initial Fig. 4e to make it much clearer (now in Fig. 7).

This referee correctly states that at first, it appears counterintuitive that « Tet1/2 and pluripotency mRNA hydroxylation is so abundant in ESC state where pluripotency factor expression is high and pluripotency is maintained ».

It is well known that specific gene expression programs in ESCs allow them to self-renew, yet they remain poised to differentiate into all cell types in response to developmental cues. To this end, key cell fate determinants need to be expressed at appropriate levels, ensuring that lineage-specific genes are adequately repressed so that orderly differentiation of ESC can occur (e.g. Smith et al. Nat Rev Mol Cell Biol 2016). For example, should pluripotency factors be expressed at too-high levels, this would lead to strong silencing of lineage-commitment genes, with cells being « stuck »/remaining in the pluripotent state (Leconte et al. Science 2015).

Our data support a working model wherein 5hmC marks key ESC fate determinants to limit their levels and ensure their continuous degradation. Specifically, 5hmC would contribute to controlling the abundance of pluripotency factors (such as Eed or Jarid2), so that they would be expressed at the appropriate levels (i.e. not too high, not too low). This would ensure adequate repression of lineage-specific factors while, crucially, preparing ESCs to rapidly respond to differentiation stimuli.

We thus believe that this model explains why Tet1/2 and hydroxylation of pluripotency mRNA need to be high in the ESC state, as our proposed stepwise mechanism highlights 5hmC mRNA modification as an essential regulatory layer, safeguarding efficient, timely, authentic downregulation of lineage-specific genes. In this manner, 5hmC can promote a fast response to external cues during cell differentiation.

Minor comments:

1- « Is dRBD generated by deleting the whole RBD domain or only the 9aa? Please explain this clearly in the text. »

We apologize for the lack of clarity in describing deletion of the RBD domain. We did delete the whole RBD domain. We have modified the text to make it clearer and have modified the scheme of RBD deletion in Fig. 3a and Supplementary Fig. 3a.

2- « Why is Tet3 not induced upon differentiation in the panels shown? It is known that Tet3 levels go up upon differentiation. Actually induction of Tet3 upon differentiation may serve as an explanation for major point 7-(2). »

This referee correctly states that Tet3 is known to be upregulated during ESC differentiation. However, this upregulation depends on the type of differentiation (spontaneous or direct) and on its timing. Specifically, we used conditions for spontaneous differentiation of ESCs to EBs at an early time (day 4), conditions under which Tet1 and Tet2 expression are decreased and Tet3 is still barely expressed (Fig. 1g). These Tet expression and differentiation conditions are in agreement with previous work (Ting et al. Molecular Neurobiology 2015). This allowed us to focus on the role played by Tet1 and Tet2 at early stages of ESC differentiation. Note that decreased levels of Tet1/Tet2 expression in EBs are well in line with our observed decrease of 5hmC levels during ESC differentiation and with our proposed model (cf. point 7). It will be interesting in the future to assess the contribution of Tet3 in later steps of differentiation, e.g. neuronal differentiation, as reported for DNA hydroxymethylation (Dawlaty, Dev. Cell 2014). We have added this information in the Main text and have modified Fig. 1g to make it clearer.

3- « 5mC vs. 5hmC in RNA: If 5mC is the substrate for DNA hydroxylation, presumably in RNA it is also the substrate. What is the significance of RNA 5C methylation vs hydroxylation? RNA stability? Does loss of TET enzymes affect overall RNA methylation? Any comments and/or data will make the study more comprehensive. »

As rightly pointed out by this reviewer, 5hmC can be formed via Tet-mediated oxidation of 5mC in RNA (Fu et al. JACS 2014, Huber et al. ChemBioChem 2015, Huang et al., Chem. Sci. 2016), as in DNA. Hence, and as requested, we have performed RNA 5mC quantification in ESC WT and TKO cells by mass spectrometry. As shown in Supplementary Fig. 2b, 5mC remained at a similar level in TKO cells, consistently with previously published data (Fu et al. JACS 2014). This observation may be due the higher level of 5mC compared to 5hmC, suggesting that only a small proportion of the 5mC undergoes hydroxylation (%m5C/%5hmC = about 1000 in HEK293T and mouse brain) (Huber et al. ChemBioChem 2015). Moreover, this observation is consistent with results previously reported in the DNA context, showing that the level of 5mC remained similar between WT and Tet TKO cells (Lu et al. Genes & Dev 2014, Verma et al. Nature Genetics 2018). Nevertheless, it is worth noting that we assessed global m5C RNA levels, so we cannot exclude that specific hydroxylated regions might harbor higher levels of 5mC upon Tet depletion. We have now added this result in the Main Text.

Regarding “*the significance of RNA 5C methylation vs hydroxylation*”, recent reports have shown that m5C is involved in regulating mRNA metabolism and early developmental processes. For example, it has been suggested that m5C is present on mRNAs in mouse ESCs and in the mouse brain (Amort et al. Genome Biology 2017). Functionally, m5C mediates mRNA nuclear export (Yang et al. Cell Research 2017) and enhances mRNA stability (Chen et al. Nature Cell Biology 2019; Yang et al. Molecular Cell 2019). Here we show that 5hmC is associated with destabilization of mRNA. This opposite role of 5hmC as compared to its precursor may suggest that RNA hydroxylation is an important post-transcriptional modification with specific functions affecting mRNA metabolism. This hypothesis is supported by our data showing that 5hmC is involved in regulating the stability of key pluripotency gene mRNAs. Our findings are also in line with our recent report establishing regulation of retroviruses through 5hmC-mediated degradation (Guallar et al Nature Genetics 2018).

We have now amended the Discussion to reflect this interesting point.

Reviewer #2

We thank this reviewer for finding that “*understanding the potential roles of Tet enzymes in RNA modifications and post-transcriptional regulation is of interest* » and for a positive critique and suggestions. However, we disagree that “*this study did not provide sufficient conceptual advance or mechanistic insights* ». Our work is not merely a confirmation of the presence of RNA hydroxymethylation in mammals. The study of 5hmC function, distribution, and biological relevance is still in its infancy. We show in the present work a previously unrecognized role of Tet-mediated RNA hydroxymethylation as a mark contributing to the transcriptome flexibility required for embryonic stem cell differentiation.

To support these claims and address the further critiques of this referee and suggestions of the other reviewers, we have included several new and important data that considerably extend and strengthen our conclusions and make our manuscript much stronger. In particular, besides providing many decisive additional controls:

1. We provide insights as to which Tets are responsible for 5hmC in RNAs in ESCs. Our novel experiments suggest that, while Tet3 seems not to be involved, both Tet1 and Tet2 appear to contribute similarly to RNA m5C oxidation and seem to have redundant functions (Fig. 3g and Supplementary Fig. 2a).
2. Tets are not responsible for all of the 5hmC in mRNA. We show that this does not seem attributable to chemical processes such as ROS (Supplementary Fig. 2f). We discuss the possibility that other enzymes might be involved in RNA hydroxymethylation.
3. By performing an unbiased analysis of RNA stability of all transcripts, we provide key novel data showing a significant increase in the stability of many transcripts in TKO ESCs as compared to WT cells (Figs 5c-e).
4. We have important new data confirming our initial stability experiments on key pluripotency transcripts. Using CRISPR knock-in ESCs for Tet2WT and Δ RBD and rescue experiments performed on TKO ESCs with Tet2WT and the Tet2 catalytic mutant, we show that Tet2 RBD is required in part for transcript destabilization and that 5hmC deposition on these transcripts facilitates their degradation (Figs 6c and 6d and Supplementary Fig. 5).

Specific comments:

1. « *Fig. 1C: first, the authors need to show the observed percentage of 5hmC at each category of genomic regions next to the expected percentage. Second, ~70% of 5hmC sites were identified to be within introns, which suggests that 5hmC is more enriched in nascent, un-spliced mRNAs. This is a consequential observation, and the authors should substantiate this by performing hMeRIP-seq with nascent or pre-existing pools of mRNAs using a metabolic labeling strategy.* »

As requested, we have updated Fig. 1d by adding the expected percentage for each category of genomic regions, and we have amended the Legend accordingly.

Regarding the significance of strong intron enrichment in 5hmC, we performed the requested metabolic labelling experiment but failed to obtain a sufficient quantity of 5hmC-immunoprecipitated RNA. However, we have used an alternative strategy: assessing the level of 5hmC by dot blotting after cell fractionation. The relevance of this strategy is based on the observation that most introns are removed from nascent mRNAs co-transcriptionally, prior to mRNA transfer to the nucleoplasm (Pandya-Jones et al. RNA 2009, Bhatt et al. Cell 2012, Pandya-Jones et al. RNA 2013). Hence, we prepared the three following RNA fractions: nascent chromatin-associated, nucleoplasmic, and cytoplasmic. As shown in Supplementary Fig. 1g, we found by 5hmC dot blotting on these RNA fractions that chromatin-associated RNAs were abundantly hydroxymethylated, suggesting that 5hmC deposition could occur cotranscriptionally, as we have previously suggested (Guallar et al. Nature Genetics, 2018). Our fractionation of cellular RNA shows that 5hmC occurs mostly on introns that are removed during pre-mRNA processing (Supplementary Fig. 1g).

The interesting observation that intronic regions of unspliced nascent pre-mRNA are rich in 5hmC warrants future study. For example, hMeRIP-Seq could be performed on the above three RNA fractions, with subsequent analysis of chronological deposition and removal of 5hmC, as has been reported for m6A (Kee et al. Genes Dev 2017).

We have now incorporated these novel results into the Main Text and Supplementary Data and have amended the Discussion.

2. « *Fig. 1b: Is the ctrl sample derived from input DNA (without IP) or mock IP using IgG? As antibodies for DNA modifications such as 5hmC have been shown to have intrinsic affinity for be short unmodified repeat sequences (e.g. GTGTGTGTGT; PMID: 29941872), it is unclear whether the putative UC-rich motif is specifically associated with 5hmC sites or related to a 5hmC antibody specific artifact in hMeRIP-seq. It will be critical for the authors to use synthetic RNAs containing the UC-rich motif with distinct RNA modification states (unmodified, 5mrC and 5hmC) to confirm specificity of 5hmC antibody in the hMeRIP-seq assay* ».

We apologize for the lack of clarity in displaying the control, which is derived from input RNA (without

IP)(now Fig. 1c). We have clarified the control used in the Legends section.

We have further validated the specificity of 5hmC pulldown by performing the requested experiment. Specifically, we carried out *in vitro* transcription (IVT) of a synthetic template containing the UC-rich motifs identified in our study (e.g. UCUCUCUCUCUCUCUCUC; UCUCUCUCUGUCUCUCUC) and harboring unmodified C, 5mC, or 5hmC (Supplementary Fig. 1a). We performed hMeRIP-qPCR with these IVT-produced transcripts as spike-in. As expected, UC-rich transcripts with modified 5hmC showed strong fold enrichment. In contrast, unmodified- or 5mC-modified transcripts showed no enrichment above background (Fig. 1a). These experiments confirm the specificity of the 5hmC antibody used in hMeRIP-Seq.

We have now included this specificity control of 5hmC antibody in Main Data (Fig. 1a), and supplementary data (Supplementary Fig. 1a), and have amended the Methods section.

3. « *Extended Fig. 1d, statistical analysis of overlap between 5hmC transcripts and pluripotency-related transcripts needs to be performed (e.g. hypergeometric test).* »

In the original manuscript, we had included in the legend of Supplementary Fig. 1e the hypergeometric test ($P < 10^{-15}$). We apologize if this was not clear, and to avoid confusion, we have now also added the *P* value in Supplementary Fig. 1e.

4. « *While catalytic domains of Tet1 can catalyze RNA hydroxymethylation in vitro, previous report indicates that only full-length Tet3 (but not Tet1/2) is enzymatically active in catalyzing 5hmC in RNAs in vitro (PMID: 25073028). Since Tet3 is expressed at very low level in mESCs, it is thus unclear whether 5hmC in mESCs is derived from low level Tet3 or low activity of Tet1/2. To dissect the underlying mechanism, the authors should use CRISPR knock-in strategy to introduce point Mutations to inactive Tet1/2/3 individually or in combination to identify which Tet enzyme(s) is responsible for the 5hmC in mESC RNAs.* »

These are pertinent comments and we value the referee's suggested strategy to use CRISPR knock-in to introduce Tet point mutations. However, this requires very demanding and tricky experiments and we have used alternative approaches which we feel, as described below, satisfactorily answer this referee's requests:

- Regarding Tet3, while it is true that a previous study using Tet3 overexpressed in HEK293 suggested full-length Tet3 as the only RNA hydroxymethyltransferase (Fu et al. JACS 2014), similar experiments have shown that all three Tets are enzymatically active toward mRNA (Xu et al. Plos One 2016). Our further analyses indicate that Tet3 seems not to be involved in 5hmC marking of mRNAs in ESCs : (i) dot blots in Tet1- and Tet2-depleted ESCs (DKO), show reduction of 5hmC levels similar to that observed in TKO ESCs (Supplementary Fig. 2a), (ii) dot blots in Tet3 KO ESCs do not show any decrease in global 5hmC (Supplementary Fig. 2a).
- Concerning Tet1 and Tet2, it would seem that both Tet1 and Tet2 contribute similarly to RNA m5C oxidation and have redundant functions. These conclusions stem from the following novel results:
 - i. We did loss-of-function experiments using CRISPR knock-in ESCs for Tet2 wt and Tet2 Δ RBD, followed by hMeRIP-Seq. We observed a significant decrease (67.5%) of 5hmC-enriched regions upon deletion of Tet2 RBD (Fig. 4 and Supplementary Table 6). This shows, interestingly, that Tet2, at least via its RBD, contributes to hydroxymethylation of mRNAs. This is in line with our recent work showing Tet2-mediated RNA hydroxymethylation of retroviruses (Guallar et al. Nature Genetics 2018).
 - ii. We further extended our comparisons of Tet1 and/or Tet2 RIP-Seq with hMeRIP data in ESCs (Fig. 3g). We found that when Tet1 and Tet2 are bound to 5hmC targets, they are mostly bound together, rather than alone (Fig. 3g). This suggest that both Tet1 and Tet2 contribute to 5hmC and that they have redundant roles in mRNA hydroxymethylation in ESCs. Future analyses will be needed to decipher the mechanisms by which Tet1 and Tet2 can substitute for one another for RNA m5C oxidation.

We have incorporated all the above results and information in the Main Text and have amended the Discussion.

5. « *Fig. 2: Why are Tet enzymes only responsible for >50% of 5hmC in RNAs? The presence of appreciable levels of 5hmC in Tet TKO ESCs (>50% remaining) suggests that other enzymes(s) might be involved in oxidizing 5mC to 5hmC in mammalian cells. Some 5hmC may also be induced by cellular reactive oxygen species (ROS). It will be very helpful if the authors can experimentally test pharmacological means to manipulate ROS pathway and test whether this pathway contributes to 5hmC in RNA.* »

We value this comment, as our data show indeed that Tet enzymes are only partly responsible for deposition of 5hmC in mRNA. This appears both from the overlap between RIP-Seq for Tet1/Tet2 and hMeRIP-seq in wt ESCs (Fig. 3g) and from mass spec and hMRerIP-seq performed on Tet-knockout ESCs (Fig. 2). Our findings are consistent with earlier reports (Huber et al. ChemBioChem 2015, Fu et al. JACS 2014).

As asked, we have tested whether 5hmC in mRNA might form through other chemical processes. Specifically, we have evaluated whether 5hmC might be induced by cellular reactive oxygen species (ROS). This seems not to be the case, as treatments of ESCs with either buthionine sulfoximine (BSO) or H₂O₂ did not change global 5hmC mRNA levels (Supplementary Fig. 2f). We cannot exclude that other chemical processes might be involved, and future work should address this possibility.

As also rightly suggested by this referee, 5hmC in RNA might be formed by enzymes other than Tets. Tets belong to the family of ferrous-ion- and α -KG-dependent dioxygenases (Fe²⁺ and 2-OG). This family includes more than 60 members, among which: (1) nucleic acid oxygenases (NAOXs), such as Tet and Alkbh enzymes (e.g. FTO); (2) hydroxylases; (3) JmjC-domain-containing enzymes, such as histone lysine demethylases. Interestingly, several of these enzymes are already known to modify various substrates (Klose et al. Nature Reviews Genetics 2006; Loenarz & Schofield Nature Chemical Biology 2008; Cloos et al. Genes Dev. 2008). Consequently, one way to start identifying novel RNA hydroxymethyltransferases might be (1) to perform *in vitro* hydroxymethylation assays on TKO-cell extracts in the presence of various co-factors (e.g. Fe²⁺, α -KG, or FAD) and under different conditions (IDH1/2 mutations, presence of a 2-OG analogue or MAO inhibitor). and (2) to perform, in TKO cells, CRISPR KO screening targeting the whole above-mentioned family, with determination of relative 5hmC levels in RNA by mass spectrometry. Such work is clearly of interest, and should be done in the future.

We have amended the Main Text and the Discussion to reflect the above interesting points.

6. « Fig. 3: Why the majority of Tet1- or Tet2-bound target RNAs do not contain 5hmC site? What is the difference between these two groups of Tet1/2 bound RNAs? The authors should further analyze the sequence context, expression level, RNA binding protein sites, other RNA modifications (e.g. 6mA) or other genomic features between these two groups of RNAs».

We appreciate these insightful comments. Regarding the questions «Why the majority of Tet1- or Tet2-bound target RNAs do not contain 5hmC site?», we believe that this suggests the interesting possibility that, besides hydroxylating mRNA, Tet1/Tet2 might also function independently of their catalytic activity. Such an «RNA hydroxymethylation independent» role would be analogous to the well-described non-catalytic action of Tet1/Tet2 on DNA. For example, genome-wide DNA hydroxymethylation (hMeDIP) and ChIP-Seq have revealed that 5hmC-positive genes represent only 35% of Tet1-binding DNA targets (Williams et al. Nature 2011). On DNA, it is also well described that, in a catalytic-independent manner, Tet proteins associate with diverse chromatin-related machineries, such as HDAC or SET1/COMPASS, involved in transcriptional repression or activation (Delatte et al. EMBO J 2014).

Tets seem likewise to have a non-catalytic action on RNA. In favor of this view, we have recently reported that a catalytic-activity-independent function of Tet2 is involved in regulating some retroviruses (Guallar et al. Nature Genetics 2018). Specifically, we have shown in mouse ES cells that endogenous retrovirus (ERV) transcripts are regulated by two mechanisms: (1) post-transcriptional silencing of ERV RNAs via Tet2-mediated RNA hydroxymethylation; (2) transcriptional repression of ERVs through binding of Tet2 to RNA and concomitant recruitment of HDAC activity.

All in all, the above observations suggest that binding of Tet1/Tet2 on mRNAs that are not hydroxylated might provide a platform for epigenetic complexes, such as HDAC, for transcriptional regulation. We now mention such catalytic-independent functions of Tet1/Tet2 on mRNAs, which is open to future work, in the Discussion.

Regarding the question «What is the difference between these two groups of Tet1/2 bound RNAs?», we analyzed several features, as requested. Of note, our Tet1/2 RIP-Seq methodology did not allow to analyze sequence context and RNA binding protein sites, as IP was performed without any prior transcript fragmentation, and we thus assessed enrichment of the whole transcript. Using publicly available data, when comparing Tet1/2-bound 5hmC-modified and unmodified RNA, we observed no significant difference looking at the following features: m6A modification, DNA hydroxymethylation, DNA methylation at promoters (data not shown). However, when we compared the level of 5-methylcytosine in gene bodies from which transcripts are bound by Tet1/2, we observed a higher level of 5mC in genes related to unmodified RNA compared to the 5hmC modified ones (Supplementary Fig. 3e). We have now included this new result in the Main Text and amended the Discussion.

7. « Fig. 4: In addition to four analyzed genes (Eed, Jarid2, Smarcc1, and Dab1), the authors should perform unbiased analysis of RNA stability of all transcripts (RNA-seq) after adding the transcription inhibitor. »

As pertinently suggested by this reviewer, we have measured mRNA stability in WT and TKO ESCs by transcriptome-wide monitoring of mRNA levels by RNA-seq after transcription inhibition with alpha-amanitin (Fig. 5c).

As depicted in Figs 5d-e (and Supplementary Table 7), we observed longer mRNA half-lives upon Tets depletion in TKO versus WT ESCs. These key unbiased analyses support and extend our initial data in favor of a role of Tet-mediated RNA hydroxymethylation in transcript stability.

We have now incorporated these results in the Main Text.

Reviewer #3

We thank the reviewer for finding that “*these discoveries are both novel and interesting*”. We also appreciate his/her insightful suggestions.

1- « *The main novel part of this manuscript was the demonstration of 5hmC in promoting mRNA decay. Nevertheless, the same group has previously reported 5hmC regulates mRNA translation, and another Chinese group has showed that m5C stabilizes mRNA and regulates zebra fish early embryogenesis and human bladder carcinogenesis. The authors are encouraged to discuss the distinct roles of m5C and 5hmC in mRNA metabolism in the Discussion section.* »

We agree with the referee that our previous findings suggesting a role of 5hmC in regulating translation in *Drosophila* differ from our novel data on mouse ESCs, where observed no difference in translation efficiency. This is an interesting finding, open to further investigation. The fact that different models were used in the two studies may explain the opposite 5hmC functions. Actually, the importance of context for RNA modification regulation and function has been emphasized previously, particularly in the case of m6A (e.g. Shi et al. *Molecular Cell Review* 2019). It appears that the occurrence and biological outcome of m6A modification at a given site on mRNA can be highly context-dependent, providing unique features associated with different biological processes. This may be due in part to m6A writers and readers, such as those of the METTL3 and YTH families, respectively, which strongly impact the fate of m6A-modified mRNA in a context-dependent manner (e.g. Xiang et al. *Nature* 2017, Wang et al. *Cell* 2015, Shi et al. *Cell Research* 2017, Xiao et al. *Mol. Cell* 2016, Shima et al. *Cell Reports* 2017).

Considering the major roles of writers and readers in determining the regulatory roles of RNA modifications, it will be crucial, in the future, to characterize 5hmC effectors, including readers, in order to better understand the context-dependent functions of this mark.

We apologize for omitting to discuss the distinct roles of 5mC and 5hmC in mRNA metabolism. Recent reports suggest that m5C is involved in regulating mRNA metabolism and early developmental processes. As revealed by high-throughput sequencing, m5C is present on mRNAs in mouse embryonic stem cells and in the mouse brain, where it may play important roles (Amort et al. *Genome Biology* 2017). Among its reported functions, m5C has recently been reported to mediate mRNA nuclear export through its binding to ALYREF (Yang et al. *Cell Research* 2017). This mark has further been shown to have a role in HIV infection through modulation of RNA splicing and translation (Courtney et al. *Cell Host & Microbe* 2019). In addition to the reported roles of 5mC in mRNA fate, this mark has interestingly been suggested to enhance mRNA stability (Warren et al. *Cell Stem Cell* 2010, Zhang et al. *Nature Communications* 2012). Recent findings have shown that 5mC-modified RNAs are stabilized through their preferential binding to YBX1. This interaction affects physiological events such as zebrafish embryo development and also pathological events such as oncogene activation in human urothelial carcinoma of the bladder (Yang et al. *Molecular Cell*, 2019, Chen et al. *Nature Cell Biology* 2019).

Here we show that 5hmC is associated with mRNA destabilization. This opposite role of 5hmC as compared to its precursor may suggest that RNA hydroxylation is an important post-transcriptional modification with specific functions affecting mRNA metabolism. This hypothesis is supported by our data showing that 5hmC is involved in regulating the stability of key pluripotency gene mRNAs. Our findings are also in line with our recent report establishing regulation of retroviruses through 5hmC-mediated degradation (Guallar et al. *Nature Genetics* 2018).

We now mention this information in the Discussion.

2- « *The authors used Tet1/2/3 triple-knockout (TKO) mouse ESCs. Did the Tet1/2/3 family proteins have redundant mRNA targets? Which one of Tet proteins plays a major role in RNA m5C oxidation?* »

This referee raises an interesting point. Our additional analyses and experiments suggest that in ESCs, while Tet3 seems not to be involved, both Tet1 and Tet2 contribute similarly to RNA 5mC oxidation and seem to have redundant functions. These conclusions stem from the following novel results:

(i) We did loss-of-function experiments using CRISPR knock-in ESCs for Tet2 wt or Δ RBD, followed by hMeRIP-Seq. Interestingly, we observed a significant decrease (67.5%) of 5hmC-enriched regions upon deletion of Tet2 RBD (Fig. 4 and Supplementary Table 6). This shows that Tet2, at least via its RBD, contributes to hydroxymethylation of mRNAs.

(ii) Tet3 seems not to be involved in 5hmC marking of mRNAs in ESCs. First, dot blots in Tet1- and Tet2-depleted (DKO) ESCs, show a reduction of 5hmC levels similar to that observed in TKO ESCs (Supplementary Fig. 2a). Second, dot blots from Tet3 KO ESCs show no decrease in global 5hmC (Supplementary Fig. 2a). This does not exclude that Tet3 might be involved in RNA hydroxymethylation in other cell contexts.

(iii) We further extended our comparisons of Tet1 and/or Tet2 RIP-Seq with hMeRIP data on ESCs (Fig. 3g). We found that when Tet1 and Tet2 are bound to 5hmC targets, they are mostly bound together, rather than alone (Fig. 3g). This suggest that both Tet1 and Tet2 contribute to 5hmC and that they have redundant roles in mRNA hydroxymethylation in ESCs. Future analyses will be needed to decipher the mechanisms by which Tet1 and Tet2 can substitute for one another for RNA 5mC oxidation.

We have incorporated all the above results and information in the Main Text and have amended the Discussion.

3- « *The authors may consider to apply 5hmC-targeted mRNA Reporter gene assays to provide supporting evidence for Tet-5hmC dependent hypothesis. If transfection is easy to be manipulated in ESC, this new piece of information will be a good addition to this work.* »

We have produced unmodified and 5hmC-modified transcripts by *in vitro* transcription in the presence of C or 5hmC nucleotides and have used them to transfect WT ESCs. Their abundance was measured 6h and 24h post-transfection in order to evaluate their relative stability. These time points were chosen according to previously reported work and are described as appropriate for studying transcript stability in ESCs (Wroblewska et al. Nature Biotechnology 2015). We observed after ESC transfection that *in vitro* 5hmC-modified transcripts were less stable than their unmodified counterparts (Figs 6a and 6b). These *in vitro* data are in good agreement with our original finding that 5hmC favors fast turnover of RNA transcripts.

This important new result is now added in the Main Text and in a brand-new Figure (Fig. 6).

4- « *The out frame of Fig. panel can be deleted.* »

We apologize if the out frame of Fig. panels made them less clear. We have now removed all panel frames as requested.

REVIEWERS' COMMENTS:

Reviewer #1 (Remarks to the Author):

The authors have addressed my comments with great detail through new experiments and analyses as well as revised the discussion and proposed model. The manuscript is very much improved and will be a nice contribution to the field. The model can still be interpreted or seen from different angles in part due to dynamics of Tet expression during ESC to EB differentiation. Hopefully future work involving directed differentiation to a specific lineage, with a focus on a defined set of lineage specific genes, can further build around this model and refine and strengthen it. For now, I commend them on their very detailed responses to my comments, and support the publication of the manuscript.

Reviewer #2 (Remarks to the Author):

The revised manuscript is much improved and addressed most of my initial concerns. I do not have additional question.

Reviewer #3 (Remarks to the Author):

The authors have addressed most of my concerns. And I recommend its publication.

Reply to the referees:

Reviewer #1 (Remarks to the Author):

The authors have addressed my comments with great detail through new experiments and analyses as well as revised the discussion and proposed model. The manuscript is very much improved and will be a nice contribution to the field. The model can still be interpreted or seen from different angles in part due to dynamics of Tet expression during ESC to EB differentiation. Hopefully future work involving directed differentiation to a specific lineage, with a focus on a defined set of lineage specific genes, can further build around this model and refine and strengthen it. For now, I commend them on their very detailed responses to my comments, and support the publication of the manuscript.

We greatly appreciate the reviewer's comments concerning the improvement of the manuscript, and we agree that future work will be valuable to strengthen and refine our proposed model.

Reviewer #2 (Remarks to the Author):

The revised manuscript is much improved and addressed most of my initial concerns. I do not have additional question.

We are thankful to the reviewer for his/her comment concerning our revised manuscript.

Reviewer #3 (Remarks to the Author):

The authors have addressed most of my concerns. And I recommend its publication.

We are thankful to the reviewer for supporting the publication of our manuscript.